# Demographic histories and genetic diversity across pinnipeds are shaped by human exploitation, ecology and life-history

M.A. Stoffel[1,2], E. Humble[1,3], A.J. Paijmans [1], K. Acevedo-Whitehouse[4], B.L. Chilvers[5], B. Dickerson[6], F. Galimberti [7], N.J. Gemmell [8], S.D. Goldsworthy[9], H.J. Nichols[2,18,19], O. Krüger[1], S. Negro[10,20], A. Osborne[11], T. Pastor[12], B.C. Robertson[13], S. Sanvito [7], J.K. Schultz[14], A.B.A. Shafer[15], J.B.W. Wolf[16,17] & J.I. Hoffman[1,3]

A central paradigm in conservation biology is that population bottlenecks reduce genetic diversity and population viability. In an era of biodiversity loss and climate change, understanding the determinants and consequences of bottlenecks is therefore an important challenge. However, as most studies focus on single species, the multitude of potential drivers and the consequences of bottlenecks remain elusive. Here, we combined genetic data from over 11,000 individuals of 30 pinniped species with demographic, ecological and life history data to evaluate the consequences of commercial exploitation by 18th and 19th century sealers. We show that around one third of these species exhibit strong signatures of recent population declines. Bottleneck strength is associated with breeding habitat and mating system variation, and together with global abundance explains much of the variation in genetic diversity across species. Overall, bottleneck intensity is unrelated to IUCN status, although the three most heavily bottlenecked species are endangered. Our study reveals an unforeseen interplay between human exploitation, animal biology, demographic declines and genetic diversity.

[1] Department of Animal Behaviour, Bielefeld University, Postfach 100131, 33501 Bielefeld, Germany. [2] School of Natural Sciences and Psychology, Faculty of Science, Liverpool John Moores University, Liverpool L3 3AF, UK. [3] British Antarctic Survey, High Cross, Madingley Road, Cambridge CB3 OET, UK. [4] Unit for Basic and Applied Microbiology, School of Natural Sciences, Autonomous University of Queretaro, Avenida de las Ciencias S/N, Queretaro 76230, Mexico. [5] Wildbase, Institute of Veterinary, Animal and Biomedical Science, Massey University, Private Bag 11222, Palmerston North 4442, New Zealand. [6] National Marine Mammal Laboratory, Alaska Fisheries Science Center, National Marine Fisheries Service, National Oceanic and Atmospheric Administration, Seattle 98115 WA, USA. [7] Elephant Seal Research Group, Sea Lion Island, FIQQ 1ZZ, Falkland Islands. [8] Department of Anatomy, University of Otago, PO Box 56, Dunedin 9054, New Zealand. [9] South Australian Research and Development Institute, West Beach, SA 5024, Australia. [10] UMR de Génétique Quantitative et Évolution – Le Moulon, INRA, Université Paris-Sud, CNRS, AgroParisTech, Université Paris-Saclay, Gif-sur-Yvette 91190, France. [11] School of Biological Sciences, University of Canterbury, Private Bag 4800, Christchurch, New Zealand 8140. [12] EUROPARC Federation, Carretera de l'Església, 92, 08017 Barcelona, Spain. [13] Department of Zoology, University of Otago, PO Box 56, Dunedin 9054, New Zealand. [14] National Marine Fisheries Service, National Oceanic and Atmospheric Administration, 1315 East West Highway, Silver Spring, MD 20910, USA. [15] Forensic Science & Environmental Life Sciences, Trent University, Peterborough, ON, Canada K9J 7B8. [16] Division of Evolutionary Biology, Faculty of Biology, LMU Munich, Planegg-Martinsried, Munich 82152, Germany. [17] Science of Life Laboratory and Department of Evolutionary Biology, Uppsala University, Uppsala 752 36, Sweden. [18]Present address: Department of Animal Behaviour Bielefeld University, Postfach 100131 33501 Bielefeld, Germany. [19]Present address: Department of Biosciences, Swansea University, Swansea SA2 8PP, UK. [20]Present address: GIGA-R, Medical Genomics - BIO3, Université de Liège, Liège 4000, Belgium. Correspondence and requests for materials should be addressed to J.I.H. (email: joseph.hoffman@uni-bielefeld.de)

Unravelling the demographic histories of species is a fundamental goal of population biology and has tremendous implications for understanding the genetic variability observed today[1,2]. Of particular interest are sharp reductions in the effective population size ($N_e$) known as population bottlenecks[3,4], which may negatively impact the viability and adaptive evolutionary potential of species through a variety of stochastic demographic processes and the loss of genetic diversity[5–8]. Specifically, small bottlenecked populations have elevated levels of inbreeding and genetic drift, which decrease genetic variability and can lead to the fixation of mildly deleterious alleles and ultimately drive a vortex of extinction[6,8–10]. Hence, investigating the bottleneck histories of wild populations and their determinants and consequences is more critical than ever before, as we live in an era where global anthropogenic alteration and destruction of natural habitats are driving species declines on an unprecedented scale[11,12].

Unfortunately, detailed information about past population declines across species is sparse because historical population size estimates are often either non-existent or highly uncertain[13,14]. A versatile solution for inferring population bottlenecks from a single sample of individuals is to compare levels of observed and expected genetic diversity, the latter of which can be simulated under virtually any demographic scenario based on the coalescent[15–17]. A variety of approaches based on this principle have been developed, one of the most widely used being the heterozygosity-excess test, which compares the heterozygosity of a panel of neutral genetic markers to the expectation in a stable population under mutation-drift equilibrium[18]. Although theoretically well grounded, these methods are highly sensitive to the assumed mutation model, which is seldom known[19]. A more sophisticated framework for inferring demographic histories is coalescent-based approximate Bayesian computation (ABC)[20]. ABC has the compelling advantages of making it possible to (i) compare virtually any demographic scenario as long as it can be simulated, (ii) estimate key parameters of the model such as the bottleneck effective population size and (iii) incorporate uncertainty in the specification of models by defining priors. Due to this flexibility, ABC has become a state of the art approach for inferring population bottlenecks as well as demographic histories in general[20–29].

Although the widespread availability of neutral molecular markers such as microsatellites has facilitated numerous genetic studies of bottlenecks in wild populations, the vast majority of studies focused exclusively on single species and were confined to testing for the presence or absence of bottlenecks. We therefore know very little about the intensity of demographic declines and how these are influenced by anthropogenic impacts as well as by factors intrinsic to a given species. For example, species occupying breeding habitats that are more accessible to humans would be expected to be at higher risk of declines, while species with highly skewed mating systems tend to have lower effective population sizes[30] and might also experience stronger demographic declines as only a fraction of individuals contribute towards the genetic makeup of subsequent generations. Consequently, to disentangle the forces shaping population bottlenecks, we need comparative studies incorporating genetic, ecological and life-history data from multiple closely related species within a consistent analytical framework.

Another question that remains elusive due to a lack of comparative studies is to what extent recent bottlenecks have impacted the genetic diversity of wild populations. While a number of influential studies of heavily bottlenecked species have indeed found very low levels of genetic variability[31–34], others have reported unexpectedly high genetic variation after supposedly strong population declines[23,35–38]. Hence, it is not yet clear how population size changes contribute towards one of the most fundamental questions in evolutionary genetics—how and why genetic diversity varies across species[2,39–41]. To tackle this question, we need to compare closely related species because deeply divergent taxa vary so profoundly in their genetic diversity due to differences in their life-history strategies that any effects caused by variation in $N_e$ will be hard to detect and decipher[40,41].

Finally, the relative contributions of genetic diversity and demographic factors towards extinction risk remain unclear. While historically there has been a debate about the immediate importance of genetic factors towards species viability[5,7], there is now growing evidence that low genetic diversity increases extinction risk[8,42] and on a broader scale that threatened species tend to show reduced diversity[7]. Nevertheless, due to a lack of studies measuring bottlenecks consistently across species, it remains an open question as to how the loss of genetic diversity caused by demographic declines ultimately translates into a species' extinction risk, which can be assessed by its International Union for Conservation of Nature (IUCN) status.

An outstanding opportunity to address these questions is provided by the pinnipeds, a clade of marine carnivores inhabiting nearly all marine environments ranging from the poles to the tropics and showing remarkable variation in their ecological and life-history adaptations[43]. Pinnipeds include some of the most extreme examples of commercial exploitation known to man, with several species including the northern elephant seal having been driven to the brink of extinction for their fur and blubber by 18th to early 20th century sealers[13]. By contrast, other pinniped species inhabiting pristine environments such as Antarctica have probably had very little contact with humans[13]. Hence, pinnipeds show large differences in their demographic histories within the highly constrained time window of commercial sealing and thereby represent a unique natural experiment for exploring the causes and consequences of recent bottlenecks.

Here, we conducted a broad-scale comparative analysis of population bottlenecks using a combination of genetic, ecological and life-history data for 30 pinniped species. We inferred the strength of historical declines across species from the genetic data using two complimentary coalescent-based approaches, heterozygosity-excess and ABC. Heterozygosity-excess was used as a measure of the relative strength of recent population declines, while a consistent ABC framework was used to evaluate the probability of each species having experienced a severe bottleneck during the known timeframe of commercial exploitation, as well as to estimate relevant model parameters. Finally, we used Bayesian phylogenetic mixed models to investigate the potential causes and consequences of past bottlenecks while controlling for phylogenetic relatedness among species. We hypothesised that (i) extreme variation in the extent to which species were exploited by man should be reflected in their genetic bottleneck signatures; (ii) ecological and life-history traits could have an impact on the strength of bottleneck signatures across species; (iii) past bottlenecks should reduce contemporary genetic diversity; and (iv) heavily bottlenecked species with reduced genetic diversity will be more likely to be of conservation concern.

We report striking variation in genetic bottleneck signatures across pinnipeds, with 11 species exhibiting strong genetic signatures of population declines and estimated bottleneck effective population sizes reflecting just a few tens of surviving individuals in the most extreme cases. Despite being caused by human exploitation, these genetic bottlenecks are mediated by both breeding habitat and mating system variation, implying that species ecology and life-history contribute towards responses to anthropogenic exploitation. Furthermore, up to five-fold variation in genetic diversity across species is explained by a

combination of bottleneck history, global abundance and breeding habitat. Finally, exploring the consequences of historical bottlenecks for conservation, we show that genetic bottleneck signatures are unrelated to IUCN status across all species, although three of the four most heavily bottlenecked species are currently endangered. We conclude that the genetic consequences of anthropogenic exploitation depend heavily on a species' biology, while quantifying demographic histories can substantially contribute to understanding patterns of genetic diversity across species.

## Results

**Genetic data.** We analysed a combination of published and newly generated microsatellite data from 30 pinniped species, with a median of 253 individuals and 14 loci per species (see Methods and Supplementary Table 1 for details). Measures of genetic diversity, standardised across datasets as the average per ten individuals, varied considerably across the pinniped phylogeny, with observed heterozygosity ($H_o$) and allelic richness ($A_r$) varying by over two and almost five-fold respectively across species (Supplementary Table 2). Both of these measures were highly correlated ($r = 0.92$) and tended to be higher in ice breeding seals, intermediate in fur seals and sea lions, and substantially lower in a handful of species including northern elephant seals and monk seals (Fig. 1a).

**Bottleneck inference.** We used two different coalescent-based approaches to infer the extent of recent population bottlenecks. First, the amount of heterozygosity-excess at selectively neutral loci such as microsatellites is an indicator of recent bottlenecks because during a population decline the number of alleles decreases faster than heterozygosity[3]. Recent bottlenecks therefore generate a transient excess of heterozygosity relative to a population at equilibrium with an equivalent number of alleles[18]. Here, we quantified the proportion of loci in heterozygosity-excess ($prop_{het-exc}$) for each species, which was highly repeatable across a range of mutation models (see Methods and Supplementary Table 3). Consequently, we focused on a two-phase model with 80% single-step mutations (TPM80), which is broadly in line with mammalian mutation model estimates from the literature[44] as well as posterior estimates from our ABC analysis (Supplementary Table 4B, Supplementary Fig. 5). Figure 1b shows a heatmap of $prop_{het-exc}$ across species, which is bounded between zero (all loci show heterozygosity-deficiency, an indicator of recent expansion) and one (all loci show heterozygosity-excess, an indicator of recent decline) whereby 0.5 is the expectation for a stable population. Considerable heterogeneity was found across species, with northern and southern elephant seals, grey seals, Guadalupe fur seals and Antarctic fur seals showing the strongest bottlenecks signals. By contrast, the majority of ice-breeding seals exhibited heterozygosity-deficiency, consistent with historical population expansions.

Second, we used ABC to select between a bottleneck and a non-bottleneck model as well as to estimate posterior distributions of relevant parameters. To optimally capture recent population size changes across species, we allowed $N_e$ to vary from pre-bottleneck to post-bottleneck in both models within realistic priors (see Methods for details) while the bottleneck model also included a severe decrease in $N_e$ to below 500 during the time of peak sealing. Therefore, both models incorporate longer-term declines or expansions within realistic bounds for all species but only the bottleneck model captures a recent and severe decrease in $N_e$ due

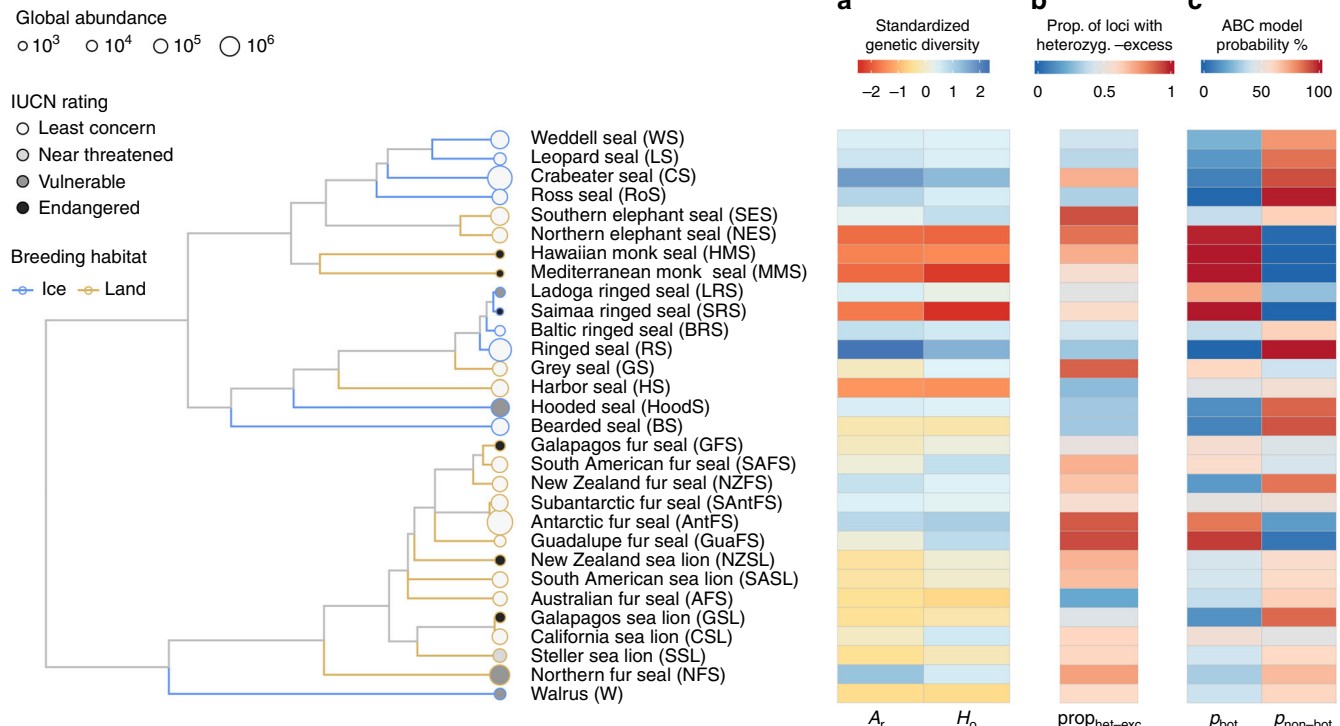

**Fig. 1** Patterns of genetic diversity and bottleneck signatures across the pinnipeds. The phylogeny shows 30 species with branches colour coded according to breeding habitat and tip points coloured and sized according to their IUCN status and global abundance respectively. **a** Shows two genetic diversity measures, allelic richness ($A_r$) and observed heterozygosity ($H_o$), which have been standardised by randomly sub-sampling ten individuals from each dataset 1000 times with replacement and calculating the corresponding mean. **b** Shows the proportion of loci in heterozygosity-excess ($prop_{het-exc}$) calculated for the TPM80 model (see Methods for details). **c** Summarises the ABC model selection results, with posterior probabilities corresponding to the bottleneck versus non-bottleneck model. The raw data are provided in Supplementary Tables 2 and 3

to anthropogenic exploitation. ABC was clearly able to distinguish between the two models, with simulations under the bottleneck model being correctly classified 85% of the time and simulations under the non-bottleneck model being correctly classified 89% of the time (Supplementary Fig. 1). A small amount of overlap between the models and therefore misclassification is unavoidable because both models were specified using broad priors to optimally fit a variety of species with vastly different population sizes. For each species, however, the preferred model showed a good fit to the observed data (all $p$-values > 0.05, Supplementary Table 5)[45]. As another indicator of model quality, posterior predictive checks[21,46] showed that the preferred models across all species were largely able to reproduce the relevant observed summary statistics (Supplementary Fig. 2). The posterior bottleneck model probability ($p_{bot}$) varied substantially across species and was strongly but imperfectly correlated with $prop_{het-exc}$ (posterior median and 95% credible intervals; $\beta = 0.17$ [0.04, 0.28], $R^2_{marginal} = 0.32$ [0.03, 0.59], see Supplementary Fig. 3). For 11 species, the bottleneck model was supported with a higher probability than the non-bottleneck model (i.e., $p_{bot} > 0.5$, see Supplementary Table 3). Subsequent parameter estimation was therefore based on the bottleneck model for eleven species and on the non-bottleneck model for the other 19 species.

Under the bottleneck model, prediction errors from the cross-validation were well below one for the bottleneck effective population size ($N_e bot$, Supplementary Table 4A and Supplementary Fig. 4) and mutation rate ($\mu$, Supplementary Table 4A) indicating that posterior estimates contain information about the underlying true parameter values. Similarly, under the non-bottleneck model, $\mu$ (Supplementary Table 4B) and the parameter describing the proportion of multi-step mutations ($GSM_{par}$, Supplementary Table 4B) were informative. By contrast, although the pre-bottleneck effective population size ($N_e hist$) also had a prediction error below one in both models, visual inspection of the cross-validation results revealed high variation in the estimates and a systematic underestimation of larger $N_e hist$ values, so this parameter was not considered further. Figure 2 shows the eleven bottlenecked species ranked in descending order of estimated posterior modal $N_e bot$ (see also Supplementary Table 4A). The parameter estimates were indicative of strong bottlenecks (i.e., $200 < N_e bot < 500$) in seven species including both phocids and otariids, while even smaller $N_e bot$ values (i.e., $N_e bot < 50$) were estimated for four phocids comprising the landlocked Saimaa ringed seal, both monk seal species and the northern elephant seal. Mutation rate estimates were remarkably consistent across species, with modes of the posterior distributions typically varying around $1 \times 10^{-4}$ (Supplementary Fig. 5 and Supplementary Table 4), while $GSM_{par}$ across species typically varied between around 0.2 and 0.3 (See Supplementary Fig. 6 and Supplementary Table 4B). Therefore, although studies of individual species are usually limited by uncertainty over the underlying mutation characteristics, our ABC analyses converged on similar estimates of mutation model and rate across species, allowing us to appropriately parameterise our bottleneck analyses.

To explore whether our results could be affected by population structure, we used STRUCTURE[47] to infer the most likely number of genetic clusters ($K$) across all datasets (see Supplementary Table 6). For all of the species for which the best supported value of $K$ was more than one ($n = 12$), we recalculated genetic summary statistics and repeated the bottleneck analyses based on individuals comprising the largest cluster. Using the largest genetic clusters did not appreciably affect our results, with repeatabilities for the genetic summary statistics and bottleneck signatures all being greater than 0.9 (see Supplementary Table

7 for repeatabilities and Supplementary Fig. 7, which is virtually identical to Fig. 1).

Furthermore, we tested all loci from each dataset for deviations from Hardy-Weinberg equilibrium (HWE, see the Methods for details). Overall, 6% of loci were found to deviate from HWE in both $\chi^2$ and exact tests after table-wide Bonferroni correction for multiple testing. To investigate whether including these loci could have affected our results, we recalculated the genetic summary statistics and repeated our bottleneck analyses after excluding them. The results remained largely unaltered, with repeatabilities all being greater than 0.97 (see Supplementary Table 8 and Supplementary Fig. 8).

Finally, we considered the possibility that our inference of recent bottlenecks could have been confounded by events further back in a species' history. In particular, increased ice cover during the last glacial maximum (LGM) could have reduced habitat availability and consequently population sizes[48–52]. We therefore tested whether small population sizes during the LGM followed by expansions could result in similar genetic patterns across pinnipeds to recent bottlenecks caused by anthropogenic exploitation (for details, see Supplementary Note 2). Specifically, we used ABC to simulate two additional demographic scenarios that were identical to the bottleneck and non-bottleneck models but which also incorporated a small population size during the LGM and subsequent expansion. ABC was not able to reliably distinguish between the two bottleneck models: correct classification rates were substantially lower at 64% for the bottleneck model and 60% for the bottleneck model incorporating post-glacial expansion. Similarly, the two non-bottleneck models had relatively poor classification rates (60% for the non-bottleneck model and 66% for the non-bottleneck model incorporating expansion). These rates are much lower than in our main analysis based on two models, indicating that ABC cannot reliably distinguish on the basis of our data between broadly equivalent models that do and do not include ice age effects. Regardless, all 11 of the species that supported the bottleneck model in the main analysis again showed the highest probability for one of the two models that incorporated a recent bottleneck (Supplementary Table 11). The fact that none of these species supported the non-bottleneck model with postglacial expansion indicates that the reduction in genetic diversity produced by a recent bottleneck can be clearly distinguished from the reduction in diversity due to a small population size at the end of the last ice age. This is to be expected as many of our summary statistics such as the M-ratio are sensitive towards recent population size changes[53].

**Factors affecting bottleneck history.** Conceivably both ecological and life-history variables could have impacted the extent to which commercial exploitation affected different pinniped species. We therefore investigated the effects of four different variables on bottleneck signatures. First, we hypothesised that breeding habitat would be important as ice-breeding species are less accessible and more widely dispersed than their land-breeding counterparts. Second, we considered sexual size dimorphism (SSD) an important life-history variable as species with a high SSD aggregate in denser breeding colonies, making them more valuable to hunters, and polygyny reduces effective population size. Third, the length of the breeding season may have impacted the vulnerability of a given species to exploitation and finally, generation time could potentially mediate population recovery. We found clear differences between ice-breeding and land-breeding seals in both $prop_{het-exc}$ and $p_{bot}$, with land-breeders on average showing stronger bottleneck signatures (Fig. 3a, b). In addition, $prop_{het-exc}$ was positively associated with SSD (Fig. 3c) but not with with $p_{bot}$ and the former relationship was robust to the exclusion of the

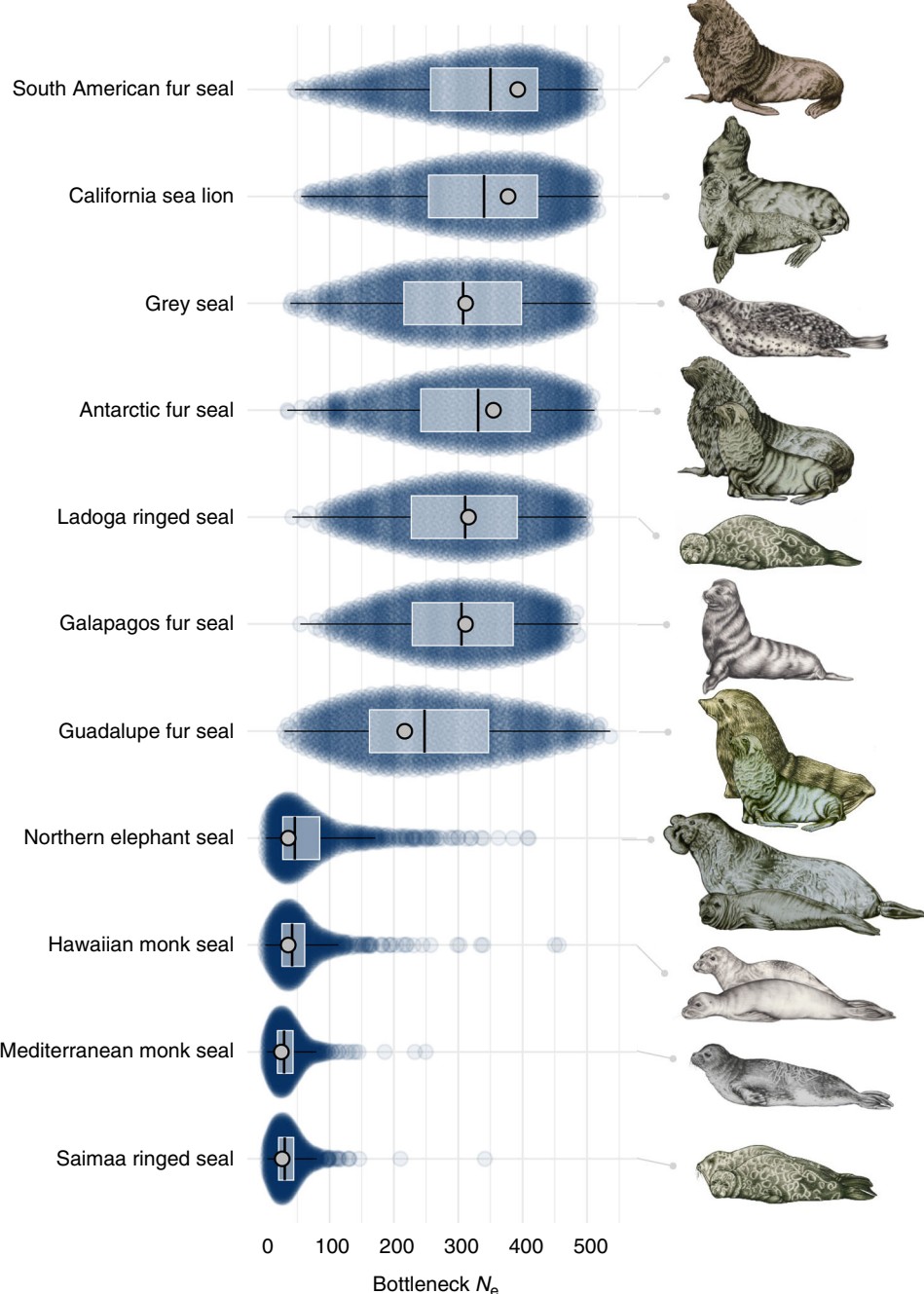

**Fig. 2** Estimated bottleneck effective population sizes. Posterior distributions of $N_e$bot are shown for 11 species for which the bottleneck model was supported in the ABC analysis, ranked according to the modes of their density distributions which reflect the estimated most likely $N_e$bot. Prior distributions are not shown as $N_e$bot was drawn from a uniform distribution with U[1, 500]. For each species, parameter values for 5000 accepted simulations are presented as a sinaplot, which arranges the data points to reflect the estimated posterior distribution. Superimposed are boxplots (centre line = median, bounds of box = 25th and 75th percentiles, upper and lower whiskers = largest and lowest value but no further than 1.5 * inter-quartile range from the hinge) with light grey points representing maximum densities. The pinniped art in this figure was created by Rebecca Carter (www. rebeccacarterart.co.uk) and is reproduced here with her permission. All rights reserved

southern elephant seal (Supplementary Fig. 9). However, we did not find the expected positive relationships with either breeding season length or generation time (see below).

To investigate this further, we constructed two Bayesian phylogenetic mixed models with prop$_{het-exc}$ and $p_{bot}$ as response variables respectively and breeding habitat, SSD, breeding season length and generation time fitted as predictors (see Methods for details). Both models explained an appreciable amount of

variation (prop$_{het-exc}$ $R^2_{marginal} = 0.58$, CI [0.22, 0.92]; $p_{bot}$ $R^2_{marginal} = 0.38$, CI [0.08, 0.62], Fig. 3d). As the four predictor variables show some level of multicollinearity (Supplementary Table 9), we reported both standardised model estimates ($\beta$) and structure coefficients ($r(\hat{Y}, x)$), which represent the correlation between each predictor and the fitted response independent of the other predictors. Breeding habitat showed the largest overall effect size in both models (Fig. 3e, Supplementary Table 10). By

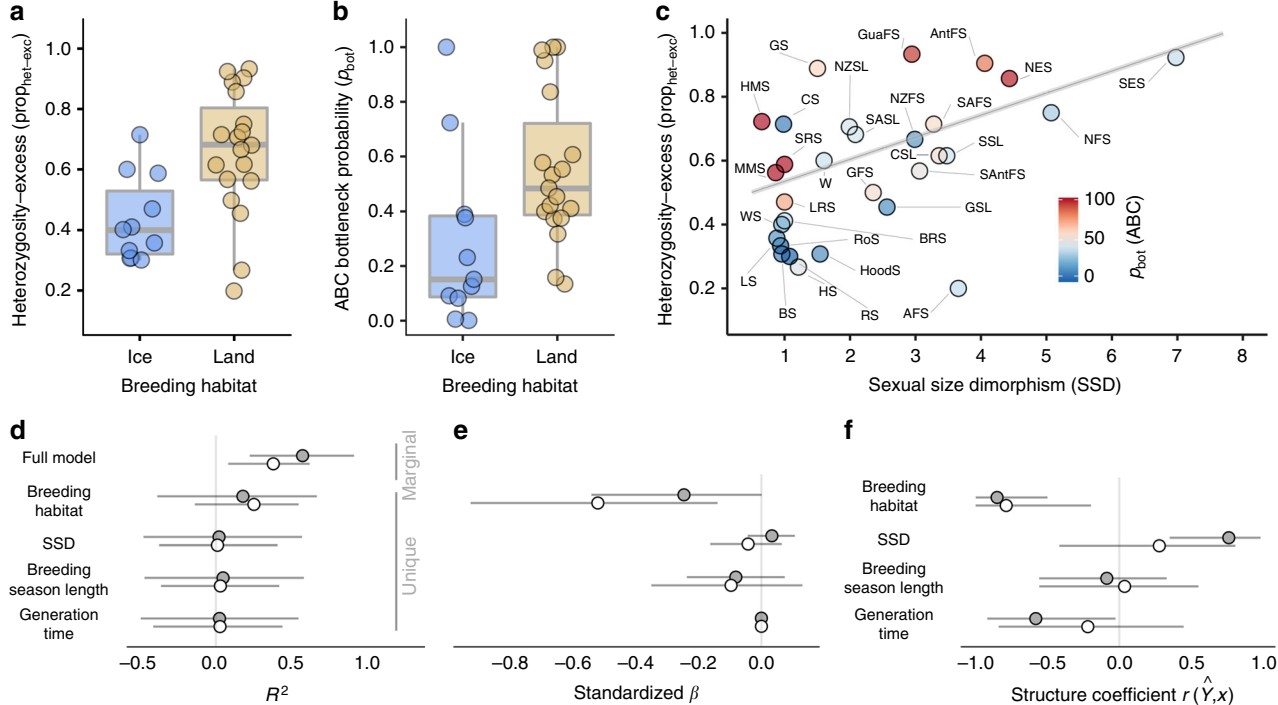

**Fig. 3** Ecological and life-history effects on bottleneck signatures. Shown are the results of phylogenetic mixed models of prop$_{het\text{-}exc}$ and $p_{bot}$ with breeding habitat, SSD, breeding season length and generation time fitted as fixed effects. **a**, **b** Show differences between ice-breeding and land-breeding species in prop$_{het\text{-}exc}$ and $p_{bot}$ respectively. Raw data points are shown together with boxplots (centre line = median, bounds of box = 25th and 75th percentiles, upper and lower whiskers = largest and smallest value but no further than 1.5 * inter-quartile range from the hinge). **c** Shows the relationship between sexual size dimorphism (SSD) and prop$_{het\text{-}exc}$, with individual points colour coded according to the ABC bottleneck probability ($p_{bot}$) and the line representing the predicted response from the prop$_{het\text{-}exc}$ model. Marginal and unique $R^2$ values, standardized $\beta$ coefficients and structure coefficients are shown for models of prop$_{het\text{-}exc}$ (filled points) and $p_{bot}$ (open points) in **d–f**, where they are presented as posterior medians with 95% credible intervals. Species abbreviations are given in Fig. 1 and Supplementary Table 1

contrast, structure coefficients showed that breeding habitat and SSD were both strongly correlated to the fitted response in the prop$_{het\text{-}exc}$ model, while SSD indeed had a much weaker effect in the $p_{bot}$ model (Fig. 3f, Supplementary Table 10). Thus, breeding habitat and SSD explain variation in prop$_{het\text{-}exc}$ whereas only breeding habitat explains variation in $p_{bot}$. We did not find a relationship between breeding season length and bottleneck signatures, with $R^2$, $\beta$ and structure coefficients all being low with broad CIs overlapping zero (Fig. 3d–f). While the structure coefficient of generation time in the prop$_{het\text{-}exc}$ model did not have CIs overlapping zero, a negative relationship is contrary to expectations and probably reflects the longer generation times of ice-breeding seals (Supplementary Fig. 10) rather than a genuine relationship.

**Determinants of genetic diversity**. To investigate the determinants of contemporary genetic diversity across pinnipeds, we constructed a phylogenetic mixed model of allelic richness ($A_r$) with log transformed global abundance, breeding habitat and SSD fitted as predictor variables together with the two bottleneck measures prop$_{het\text{-}exc}$ and $p_{bot}$ (Fig. 4). In order to avoid over-fitting the model, we did not include breeding season length and generation time, as these variables were not individually associated with $A_r$ (breeding season: $\beta = 0.01$ CI $[-0.03, 0.01]$, generation time: $\beta = 0.00$ CI $[0.00, 0.01]$). A substantial 75% of the total variation in $A_r$ was explained (Fig. 4c, $R^2_{marginal} = 0.75$, CI $[0.52, 0.91]$). Specifically, $A_r$ decreased nearly five-fold from the species with the lowest $p_{bot}$ to the species with the highest $p_{bot}$ ($\beta = -1.80$, CI $[-3.10, -0.42]$ Fig. 4a), increased by nearly five-fold

from the least to the most abundant species ($\beta = 1.38$, CI $[0.21, 2.47]$, Fig. 4b), and was on average 27% higher in ice than in land-breeding seals ($\beta = 1.76$, CI $[0.10, 3.14]$, Fig. 4b). Due to multi-collinearity among the five predictor variables (Supplementary Table 9), standardized $\beta$ estimates (Fig. 4d) can be hard to interpret because of potential suppression effects[54]. This is reflected by the low unique $R^2$ values of the predictors relative to the marginal $R^2$ of the full model (Fig. 4c). However, the structure coefficients (Fig. 4e) also revealed strong associations between the fitted model response and breeding habitat ($(r(\hat{Y}, x) = 0.54$, CI $[0.20, 0.76]$), abundance ($r(\hat{Y}, x) = 0.73$, CI $[0.54, 0.91]$) and $p_{bot}$ ($r(\hat{Y}, x) = -0.78$, CI $[10.91, -0.62]$) indicating that all three variables are associated with the response.

**Conservation status, bottleneck signatures and genetic diversity**. To investigate whether population bottlenecks and low genetic diversity are detrimental to species viability, we asked whether contemporary conservation status is related to the strength of past bottlenecks and $A_r$. Based on data from the IUCN red list (http://www.iucnredlist.org/, 2017), we classified species into two categories; the first of these, which we termed 'low concern' comprised species listed as 'least concern' and 'near threatened', while the second combined species listed as 'vulnerable' or 'endangered' into a 'high concern' category. Using a phylogenetic mixed model, we did not find any clear differences in either heterozygosity-excess or $p_{bot}$ with respect to conservation status (Fig. 5a, b). By contrast, average $A_r$ was around 1.2 alleles lower in the 'high concern' category, although there was considerable uncertainty with the

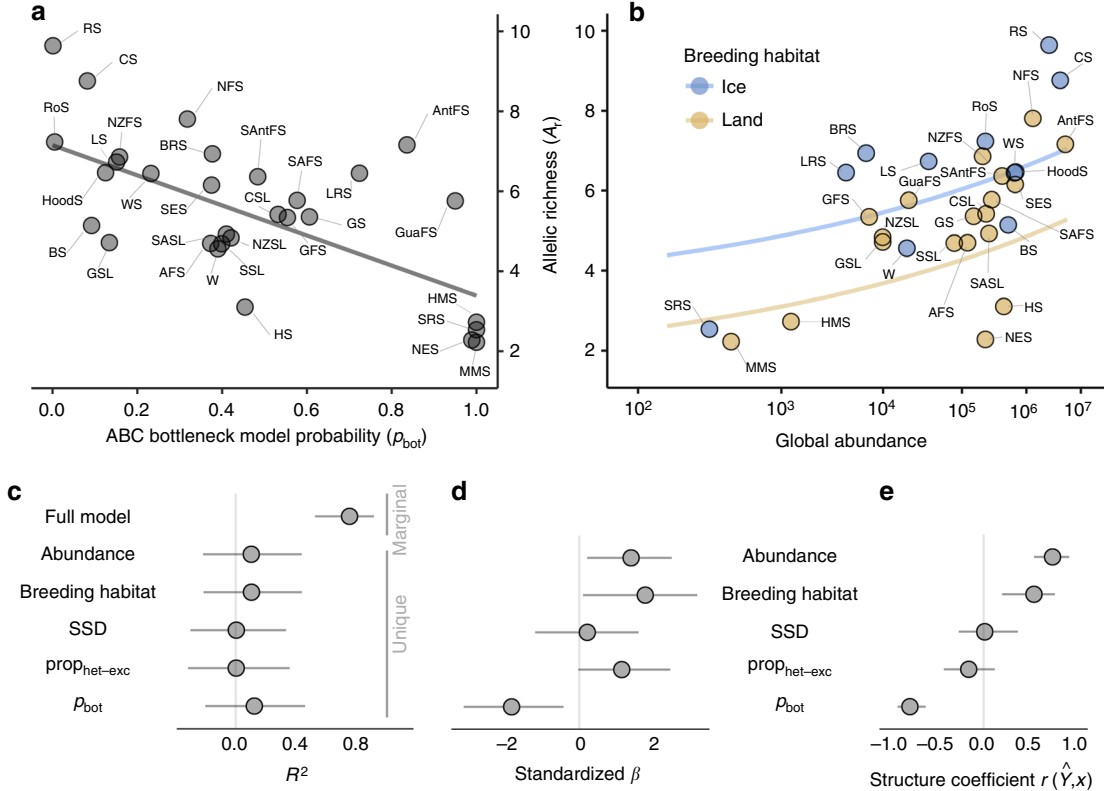

**Fig. 4** Determinants of contemporary genetic diversity across pinnipeds. **a** Shows a scatterplot of $A_r$ versus $p_{bot}$ with the grey line representing the model prediction. **b** Shows the relationship between global abundance and allelic richness ($A_r$) with the blue and yellow lines representing model predictions for ice-breeding and land-breeding seals respectively. Marginal and unique $R^2$ values, standardised $\beta$ estimates and structure coefficients for the model are shown respectively in **c–e**, where they are presented as posterior medians with 95% credible intervals. Species abbreviations are given in Fig. 1 and Supplementary Table 1

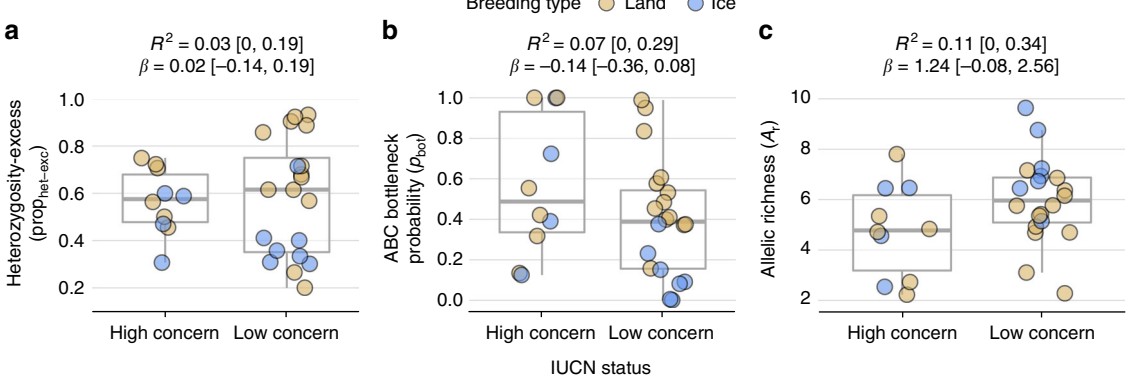

**Fig. 5** Conservation implications of bottlenecks and genetic diversity. All pinniped species were classified into either a 'low concern' or a 'high concern' category depending on their current IUCN status as described in the main text. Shown are the raw data for each category together with boxplots (centre line = median, bounds of box = 25th and 75th percentiles, upper and lower whiskers = largest and smallest value but no further than 1.5 * inter-quartile range from the hinge) for **a** $prop_{het-exc}$, **b** $p_{bot}$ and **c** $A_r$. Marginal $R^2$ and standardised $\beta$ estimates are shown for Bayesian phylogenetic mixed models with standardized predictors (see Methods for details)

95% credible interval of $\beta$ ranging from −0.08 to 2.56 (Fig. 5c).

## Discussion

To explore the interplay between historical demography, ecological and life-history variation, genetic diversity and conservation status, we used a comparative approach based on genetic data from over 80% of all extant pinniped species. To

model bottleneck strength, we used two approaches that capture different but complementary facets of genetic diversity resulting from population bottlenecks. Using ABC, we contrasted a bottleneck model incorporating a severe decrease in $N_e$ during the time of peak sealing in the 18th and 19th centuries with a non-bottleneck model. The resulting bottleneck measure, $p_{bot}$ is the probability (relative to the non-bottleneck model) that a species' observed genetic diversity is similar to the diversity of a population that experienced a severe reduction in

$N_e$ below 500, and therefore provides an absolute bottleneck measure. By contrast, heterozygosity excess (prop$_{het-exc}$) theoretically captures sudden recent reductions in $N_e$ in fairly large populations[18] and therefore provides a relative bottleneck measure. Concretely, given the average sample size of individuals and loci used in this study, we would expect to detect an excess of heterozygosity at the majority of loci (i.e., prop$_{het-exc}$ > 0.5) when a 100– to 1000–fold reduction in $N_e$ occurred, regardless of the magnitude of $N_e$ (see simulations in ref. [18]).

We specifically focused on two simple ABC models reflecting only recent demographic histories to test a clear hypothesis—large scale commercial exploitation caused severe bottlenecks and reduced the genetic diversity of many pinnipeds. This focus on a short time-frame and well known sealing history allowed us to clearly define our models around reasonable priors. Furthermore, although the genetic diversity simulated based on models of recent demographic history could in principle also be generated by more ancient bottlenecks, these are unlikely to be detected reliably using microsatellite data when a subsequent recovery occurred[55].

ABC analysis supported the bottleneck model for more than a third of the species. The strongest bottlenecks ($N_e$bot < 50) were inferred for the northern elephant seal, a textbook example of a species that bounced back from the brink of extinction[56], as well as for the two monk seals and the Saimaa ringed seal, species with very small geographic ranges and a long history of anthropogenic interaction[13]. Slightly weaker bottlenecks were estimated for seven further species including Antarctic and Guadalupe fur seals, both of which share a known history of commercial exploitation for their fur[13]. At the other end of the continuum, several Antarctic species that have not been commercially hunted such as crabeater and Weddell seals showed unequivocal support for the non-bottleneck model in line with expectations. Surprisingly, several otariid species known to have been hunted in the hundreds of thousands (e.g., South American sea lions) to millions (e.g., northern fur seals) did not show support for a bottleneck as strong as simulated in our analyses. This suggests that sufficiently large numbers of individuals must have survived despite extensive sealing, possibly on inaccessible shores or remote islands[57].

A number of factors could potentially impact our inference of the strength of recent bottlenecks across pinnipeds. First of all, population structure and deviations from HWE can affect population genetic inference. However, we found that our measures of genetic diversity as well as bottleneck signatures were highly consistent when we repeated our analyses using the largest genetic clusters or after removing loci that were out of HWE. Second, demographic events deeper in a species' history could potentially confound our inference of recent bottlenecks. However, we believe this is unlikely given the results of our supplementary analysis of postglacial expansion models and the fact that we chose our summary statistics including the M-ratio to be informative about recent population size changes. Importantly, all 11 species showing strong signatures of recent bottlenecks in our main analysis did so regardless of whether these bottlenecks were preceded by reduced population sizes followed by expansions towards the end of the late Pleistocene. Moreover, for these species, models incorporating small population sizes during the LGM did not explain the observed genetic variation better than a recent bottleneck model. A third possibility, which will affect any demographic reconstruction from genetic data, is that some of the genetic markers could be linked to loci under selection. In this case, selection would have to operate in the same direction across multiple loci within species and across species to explain our comparative patterns. However, it is not necessary to invoke selection to explain the broad-scale patterns we found across pinnipeds.

We hypothesised that not all pinniped species were equally affected by commercial exploitation partly due to intrinsic differences relating to a species' ecology and life-history. In line with this, we found a strong influence of breeding habitat on bottleneck signatures, with both prop$_{het-exc}$ and $p_{bot}$ being higher in species that breed on land relative to those breeding on ice. A likely reason for this is that terrestrially breeding pinniped species were more profitable due to their generally higher population densities and accessibility, and therefore probably experienced more intense hunting. We also found that heterozygosity-excess was strongly linked to SSD, with highly polygynous species like elephant seals and some fur seals showing the strongest footprints of recent decline. While this could reflect the increased ease of exploitation and thus higher commercial value of species that predictably aggregate in very large numbers to breed, species with higher SSD also have highly skewed mating systems making them potentially more vulnerable to severe decreases in $N_e$ when key males are taken out of the system. By contrast, we did not find an effect of SSD on the ABC bottleneck probability $p_{bot}$, suggesting that although sexually dimorphic species experienced the greatest declines, these were not necessarily as severe as simulated in the ABC analysis ($N_e$ < 500). This is probably because many species reached economic extinction well above this threshold, when populations became too small to sustain the sealing industry.

Although vast numbers of species are declining globally at unprecedented rates[12] we still lack a clear understanding of how recent declines in $N_e$ affect contemporary genetic diversity in wild populations[2,40]. Here, we explained a large proportion of the five-fold variation in allelic richness ($A_r$) observed from the most to the least diverse pinniped species. First, $A_r$ was strongly associated with $p_{bot}$ but not with prop$_{het-exc}$, in agreement with the theoretical expectation that populations have to decline to a very small $N_e$, as was simulated in our ABC analysis, to lose a substantial proportion of their diversity[3]. Second, we showed that global abundance across species was tightly linked to $A_r$ despite the likely impact of bottlenecks and the limited time-window for the recovery of genetic diversity. As differences in genetic diversity across species are largely determined by long-term $N_e$,[2] this implies that contemporary population sizes across pinnipeds must to some extent resemble patterns of historical abundance, and hence that many bottlenecked species have to a large extent rebounded to occupy their original niches. Third, $A_r$ was higher in ice-breeding relative to land-breeding seals. However, a low unique $R^2$ of breeding habitat in our model suggests that this probably reflects the more intense bottleneck histories of land-breeding seals rather than a true ecological effect.

Finally, we compared genetic diversity and bottleneck strength between species that are currently classified by the IUCN as being of conservation concern versus those that are not. We found that $A_r$ was on average around 21% lower in species within the 'high concern' category, consistent with previous evidence from a broad range of species[7]. While three out of the four pinniped species with the strongest estimated bottlenecks are currently listed as endangered, species from both categories did not overall differ in their bottleneck signatures. Our comparative study of population bottlenecks is therefore encouraging: population bottlenecks do not necessarily result in reduced genetic diversity and population viability. As shown here, global bans on commercial sealing at the beginning of the 20th century allowed many surviving pinniped populations to recover in abundance. Those that have not sufficiently rebounded illustrate the two fundamental conservation challenges, especially as biodiversity loss and climate change continue at unprecedented rates: halting population declines and promoting population recovery.

## Methods

**Genetic data.** We obtained microsatellite data for a total of 30 pinniped species including three subspecies of ringed seal (summarised in Supplementary Table 1). First, we conducted systematic literature searches to identify previously published microsatellite datasets for 25 species (see Supplementary Note 3 for details). Second, we generated new data for five species (see Supplementary Note 3 for details). Sample sizes of individuals ranged between 16 for the Ladoga ringed seal to 2386 for the Hawaiian monk seal, with a median of 253 individuals. The number of loci genotyped varied between five and 35 with a median of 14.

**Phylogenetic, demographic, life history and conservation status data.** Phylogenetic data were downloaded from the 10k trees website[58] and plotted using ggtree[59]. The three ringed seal subspecies were added according to their separation after the last ice age[60]. Demographic and life-history data for each species were obtained from refs. [61,62]. While most data stayed untransformed, we calculated SSD as the ratio of male to female body mass, and log-transformed abundance across species to account for the several orders of magnitude differences in population sizes. Data on conservation status were retrieved from the IUCN website (http://www.iucnredlist.org/, 2017).

**Data cleaning and preliminary population genetic analyses.** In order to maximise data quality, we checked all datasets by eye and generated summary statistics and tables of allele counts to identify potentially erroneous genotypes including typographical or formatting errors. In ambiguous cases, we contacted the authors to verify the correct genotypes. As several of the datasets included samples from more than one geographical location, we used a Bayesian approach implemented in STRUCTURE version 2.3.4[47] to infer the most likely number of genetic clusters ($K$) across all datasets. For computational and practical reasons, we used the ParallelStructure package in R[63] to run these analyses on a computer cluster. For all of the species for which the best supported value of $K$ was more than one, we recalculated genetic summary statistics and repeated the bottleneck analyses based on individuals comprising the largest cluster and calculated repeatabilities including 95% confidence intervals (CIs) for all variables using the rptGaussian function in the rptR package[64]. We also tested all loci from each dataset for deviations from HWE using $\chi^2$ and exact tests implemented in pegas[65] and applied Bonferroni correction to the resulting $p$-values.

**Genetic diversity statistics.** In order to examine patterns of genetic diversity across species, we calculated observed heterozygosity ($H_o$) and allelic richness ($A_r$) with strataG[66] as well as the proportion of low frequency alleles (LFA), defined as alleles with a frequency of <5%, using self-written code. For maximal comparability across species with different sample sizes, we randomly sampled ten individuals from each sample 1000 times with replacement and calculated the corresponding mean and 95% CI for each summary statistic. We did not attempt to standardise our genetic diversity measures by the number of microsatellites, as differences in the number of loci are not expected to systematically bias the mean of any summary statistic across loci.

**Heterozygosity-excess.** We quantified heterozygosity-excess using the approach of Cornuet and Luikart[18] implemented in the program BOTTLENECK version 1.2.02[67,68]. BOTTLENECK compares the heterozygosity of a locus in an empirical sample to the heterozygosity expected in a population under mutation-drift equilibrium with the same number of alleles as simulated under the coalescent[15,16]. Microsatellites evolve mainly by gaining or losing a single repeat unit[57] (the Stepwise Mutation Model, SMM), but occasional larger jump mutations of several repeat units also occur[69]. Consequently, BOTTLENECK allows the user to specify a range of mutation models, from the strict SMM through two phase models (TPMs) with varying proportions of multi-step mutations to the infinite alleles model (IAM) where every new mutation is novel. We therefore evaluated the SMM plus three TPMs with 70%, 80% and 90% single-step mutations respectively and the default variance of the geometric distribution (0.30). For each of the mutational models, the heterozygosity of each locus expected under mutation-drift equilibrium given the observed number of alleles ($H_{eq}$) was determined using 10,000 coalescent simulations. The proportion of loci for which $H_e$ was greater than $H_{eq}$ (prop$_{het-exc}$) was then quantified for all of the mutation models. To quantify consistency of the measure across mutation models, we calculated the repeatability of prop$_{het-exc}$ using the rptR package[64] in R with 1000 bootstraps while adjusting for the mutation model as a fixed effect. Although the relative pattern across species was very consistent across mutation models (repeatability = 0.81, CI = [0.71, 0.89]), absolute values of prop$_{het-exc}$ within species decreased with lower proportions of multistep mutations (means for the TPM70, 80, 90 and SMM were 0.63, 0.58, 0.49 and 0.27, respectively). Given our posterior estimates (Supplementary Fig. 6) and in line with previous studies, we therefore based our subsequent analyses on prop$_{het-exc}$ from the intermediate TPM80 model.

**Demographic models.** As a second route to inferring historical population declines, we contrasted two alternative demographic scenarios (Fig. 6) using a coalescent-based ABC framework[15,20,70,71]. To address the hypothesis that commercial exploitation from the 18th to the beginning of the 20th century led to

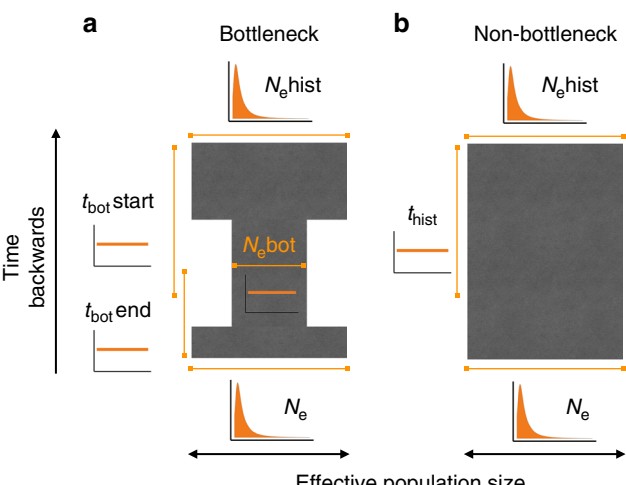

**Fig. 6** Schematic representation of two contrasting demographic scenarios. All priors were drawn independently from each other, so the current $N_e$ can be smaller or larger than $N_e$hist for a given species. This allowed both models to capture pre-bottleneck to post-bottleneck variation in population size. While $N_e$ and $N_e$hist were drawn from lognormal priors, all other parameters were specified using uniform priors. All prior distributions are also shown as small figures next to the respective parameter. The exact priors and the mutation model are given in the Methods

population bottlenecks, we first defined a bottleneck model, which incorporated a severe reduction in population size within strictly bound time priors reflecting the respective time period. This model also allowed us to capture realistic changes from the pre-bottleneck to post-bottleneck effective population size as both priors were drawn independently from the same distribution. Therefore, the model incorporates not only the bottleneck, but also longer term declines or expansions within realistic bounds as described below. For comparison, we defined a model that did not contain a bottleneck but which was identical in all other respects, which we called the non-bottleneck model. This model still allowed the population size to vary over time within a defined set of priors and thus captures realistic longer term variation in population size, but it does not include a severe recent bottleneck due to human exploitation.

Genetic data under both models were simulated from broad enough prior distributions to fit all 30 species while keeping the priors as tightly bound as possible around plausible values. The bottleneck model was defined with seven different parameters (Fig. 6a). The current effective population size $N_e$ and the historical (i.e., pre-bottleneck) effective population size $N_e$hist were drawn from a log-normal distribution with $N_e$ ~ lognorm[logmean = 10.5, logsd = 1] and $N_e$hist ~ lognorm[logmean = 10.5, logsd = 1]. This concentrated sampling within plausible ranges that fitted most species (i.e., with effective population sizes ranging from thousands to tens of thousands of individuals) while also occasionally drawing samples in the hundreds of thousands to fit the few species with very large populations. The bottleneck effective population size $N_e$bot was drawn from a uniform distribution between 1 and 500 ($N_e$bot ~ U[1, 500]) while the bottleneck start and end times $t_{bot}$start and $t_{bot}$end were drawn from uniform distributions ranging between ten and 70 ($t_{bot}$start ~ U[10, 70]) and one and 30 ($t_{bot}$end ~ U[1, 30]) generations ago respectively. Hence, the bottleneck time priors encompassed the last four centuries for all species, as their estimated generation times vary between approximately 7 and 19 years (Supplementary Table 1). The microsatellite mutation rate $\mu$ was refined after initial exploration and drawn from a uniform prior with $\mu$ ~ U[$10^{-5}$, $10^{-4}$] which lies within the range of current empirical estimates[44,72]. The mutation model was defined as a generalized stepwise mutation model with the geometric parameter GSM$_{par}$ reflecting the proportion of multistep mutations, uniformly distributed from GSM$_{par}$ ~ U[0, 0.3]. The non-bottleneck model was defined with five parameters (Fig. 6b). $N_e$, $N_e$hist, $\mu$ and GSM$_{par}$ were specified with the same priors as previously defined for the bottleneck model and the time parameter corresponding to the historical population size $t_{hist}$ was drawn from a uniform distribution ranging between 10 and 70 generations ago ($t_{hist}$ ~ U[10, 70]). All population size changes were therfore modeled as instantaneous changes at times $t_{bot}$start, $t_{bot}$end or $t_{hist}$.

**ABC analysis.** We simulated a total of $2 \times 10^7$ datasets of 40 individuals and ten microsatellite loci each under the two demographic scenarios using the fastsimcoal function in strataG[66] as an R interface to fastsimcoal2[73], a continuous-time coalescent simulator. For both the simulated and empirical data, we used five different summary statistics for the ABC inference, all calculated as the mean across loci. Allelic richness (number of alleles), allelic size range, expected heterozygosity (i.e.,

Nei's gene diversity[74]), the M-ratio[53] and the proportion of low frequency alleles (LFA) (i.e., with frequencies < 5%). The summary statistics for the empirical datasets were computed by repeatedly re-sampling 40 individuals with replacement from the full datasets and calculating the mean across 1000 subsamples (for the Ladoga ringed seal and the Baltic ringed seal which had sample sizes smaller than 40, the full datasets were taken). As a small number of loci in the empirical data exhibited slight deviations from constant repeat patterns (i.e., not all of the alleles within a locus conformed to a perfect two, three or four bp periodicity), we calculated the M-ratio as an approximation using the most common repeat pattern of a locus to calculate the range of the allele size $r$ and subsequently the $M$-ratio with $M = k/(r + 1)$ where $k$ is the number of alleles. All statistics were calculated using a combination of functions from the strataG package and self-written code. For the ABC analysis, we used a tolerance threshold of $5 \times 10^{-4}$, thereby retaining 5000 simulations with summary statistics closest to those of each empirical dataset. For estimating the posterior probability for each scenario and each species, we used the multinomial regression method[20,75] as implemented in the function postpr in the abc package[25] where the model indicator is the response variable of a polychotomous regression and the accepted summary statistics are the predictors. To construct posterior distributions from the accepted summary statistics for the model parameters, we used a local linear regression approach[20] implemented in the abc function of the abc package.

**Evaluation of model specification and model fit**. We evaluated whether ABC could distinguish between the two models by performing a leave-one-out cross validation implemented by the cv4postpr function of the abc package. Here, the summary statistics of one of the existing $2 \times 10^7$ simulations were considered as pseudo-observed data and classified into either the bottleneck or the non-bottleneck model using all of the remaining simulations. If the summary statistics are able to discriminate between the models, a large posterior probability should be assigned to the model that generated the pseudo-observed dataset. This was repeated 100 times and the resulting posterior probabilities for a given model were averaged to derive the rate of misclassification. We furthermore used a hypothesis test based on the prior predictive distribution[45] implemented in the gfit function in the abc package to check for each species that the preferred model provided a good fit to the observed data. Specifically, we used the median distance between the accepted and observed summary statistics as a test statistic, whereby the null distribution was generated using summary statistics from the pseudo-observed datasets. Hence, a non-significant $p$-value indicates that the distance between the observed summary statistics and the accepted summary statistics is not larger than the expectation based on pseudo-observed data sets, i.e., the assigned model provides a good fit to the observed data.

**Evaluation of the accuracy of parameter estimates**. In order to determine which parameters (i.e., population sizes, times and mutation rates and models) could be reliably estimated, we used leave-one-out cross validation implemented in the cv4abc function from the abc package to determine the accuracy of our ABC parameter estimates. For a randomly selected pseudo-observed dataset, parameters were estimated via ABC based on the remaining simulations using the rejection algorithm and a prediction error was calculated. This is possible because we know the true parameter values from which a given pseudo-observed dataset was simulated. This procedure was repeated 1000 times and a mean prediction error ranging between 0 and 1 was calculated, where 0 reflects perfect estimation and 1 means that the posterior estimate does not contain any information about the true parameter value[25].

**Posterior predictive checks**. To further confirm the fit of the preferred models, we conducted posterior predictive checks[21,46] for each species. First, we estimated the posterior distribution of each parameter using ABC. Second, we sampled 1000 multivariate parameters from their respective posterior distributions and used those to simulate summary statistics a posteriori based on the preferred model. Last, we plotted those summary statistics as histograms and superimposed the observed summary statistics across all species[21].

**Bayesian phylogenetic mixed models**. Finally, we used Bayesian phylogenetic mixed models in MCMCglmm[76] to evaluate the ecological and life-history variables affecting bottleneck strength and genetic diversity, and to test whether bottleneck history and genetic diversity are predictive of contemporary conservation status. Details of all the models are given in Supplementary Table 10. All of the response variables were modelled with Gaussian distributions, while the predictors were fitted as fixed effects and the phylogenetic covariance matrix as a random effect. Predictors in models containing binary fixed effects were standardised by two standard deviations to allow a direct comparison between the effect sizes[77,78]. In models without binary fixed effects, the predictor variables were standardised by one standard deviation. For all models, we report the marginal $R^2$ as in ref. [79]. Some of the predictors in our models were correlated and multicollinearity might lead to suppression effects and make the interpretation of regression coefficients difficult[54]. We therefore reported standardized $\beta$ estimates, structure coefficients, $r(\hat{Y}, x)$ and unique $R^2$ values for all variables in all models. The structure coefficients represent the correlation between a predictor and the fitted response of a model independent of the other predictors, and therefore reflect the direct contribution of a variable to that model. On the other hand, the unique $R^2$ is the difference between the marginal $R^2$ of a model including and a model excluding a predictor, which will be small when another predictor explains much of the same variation in the response[54]. All model estimates were presented as the posterior median and 95% credible intervals (CIs). We used uninformative priors with a belief (shape) parameter $v = 1$ for the variance-covariance matrices of the random effects and inverse-Wishart priors with $v = 0.002$ for residual variances. For each model, three independent MCMC chains were run for 110,000 iterations, with a burn-in of 10,000 iterations and a thinning interval of 100 iterations. Convergence was checked visually and by applying the Gelman–Rubin criterion to three independent chains. All of the upper 95% confidence limits of the potential scale inflation factors were below 1.05.

**Code availability**. All data wrangling steps and statistical analyses except for the heterozygosity-excess tests[67] were implemented in R[80]. The complete documented analysis pipeline can be downloaded from our GitHub repository https://github.com/mastoffel/pinniped_bottlenecks.

## Data availability

We provide all raw and processed data to reproduce the analyses in the paper. These can also be downloaded from our GitHub repository: https://github.com/mastoffel/pinniped_bottlenecks.

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

## Acknowledgements

This research was supported by standard grants (HO 5122/3-1 and HO 5122/5-1) from the German Research Foundation (DFG) and as part of the SFB TRR 212 (NC³, project

A01) together with a dual PhD studentship from Liverpool John Moores University. We are grateful to John Arnould, David Coltman, Corey Davis, Larissa Rosa de Oliveira, Tom Gelatt and NOAA, Melissa Gladstone, Roger Kirkwood, Melanie Lancaster, Tim Malloy, Rolf Ream, Garry Stenson and Rob Stewart for providing access to published microsatellite datasets. We also thank Matthias Galipaud for advice on statistical analysis, Kevin Arbuckle for advice on Bayesian phylogenetic mixed models and Luke Eberhart-Phillips for helpful comments on the figures. We are very grateful to Rebecca Carter (www.rebeccacarterart.co.uk) for the time and effort she dedicated to producing the pinniped illustrations. Finally, we thank the Okinawa Institute of Science and Technology Preprint Journal Club for their very helpful comments on the manuscript.

## Author contributions

Conceived the study: J.I.H. and M.A.S. Generated data: K.A., B.L.C., B.D., F.G., N.G., S.D.G., H.J.N., O.K., S.N., A.O., A.J.P., T.P., B.C.R., S.S., J.S., A.B.A.S., J.B.W.W., J.I.H. Analysed data: M.A.S., E.H. and J.I.H. Wrote the paper: M.A.S. and J.I.H. All of the authors commented upon and approved the final manuscript.

## Additional information

**Competing interests:** The authors declare no competing interests.



