## [Peer Review File · Nature Communications]

Reviewers' comments:

Reviewer #1 (Remarks to the Author):

The ms "Recent demographic histories and genetic diversity across pinnipeds are shaped by anthropogenic interactions and mediated by ecology and life-history" evaluates the evidence for genetic bottlenecks in multiple pinniped populations worldwide and tests how genetic diversity and bottleneck signal correlate with some other factors. The authors find some evidence of genetic bottlenecks but not to the extent expected given the historical exploitation of many pinnipeds. This is a well-written ms and I commend the reviewers for making the data and R script available. I have a few specific comments and questions which I detail below.

Regarding the genetic data analysed, the authors do not explain how the relevant published microsatellite data were identified or selected from the available information. I know of at least 1 source which is not included and that provides data for one of the studied species in different sampling areas (see citation below). The authors need to explain how they identified sources and, if relevant, why some were excluded (and whether this could affect results). This is important to be able to understand how representative the results may be. González-Suárez, M; Aurióles-Gamboá, D; Gerber LR (2010) Past exploitation of California sea lions did not lead to a genetic bottleneck in the Gulf of California *Ciencias Marinas* 36: 199-211. Note that the data for this study are freely available at Dryad Digital Repository doi:10.5061/dryad.1454.

Related to this issue, I was unclear on how many sampling sites were represented in each dataset. This is important to understand how representative the data are and also key to better understand the STRUCTURE analyses.

For several species the posterior probabilities for the bottleneck and neutral models are very similar (eg Walrus, Baltic ringed seal, NZ sea lion, and others). Given the posterior probability of correct model classification are relatively low (error rates of 25-30%) I am not convinced an arbitrary threshold of 0.5 can be used as evidence of fitting a bottleneck vs neutral model. Wouldn't the rejection of the neutral model be a better way to determine which species show patterns consistent with a bottleneck? Using that approach one can clearly identify the four species for which bottleneck N_e is clearly estimated as small in figure 2.

Related to this in line 197 "ABC was clearly able to distinguish between the two models, with the posterior probability of correct model classification being 75% for the bottleneck model and 71% for the neutral model" I may be misinterpreting these results, but those probabilities of correct classification do not seem particularly good, you can be wrong about 25-30% of the times, which is not trivial.

Figure 1 (and in fact all figures) is visually appealing but I find it difficult to see the probability values with the colors. Using the actual values would be much more informative and given the colours are already in a table format, that seems very much doable.

Line 199 "for every species the preferred model showed a good fit to the data" for the Saimaa ringed seal the p-value is 0.08, perhaps works discussing a bit more uncertainty in this case?

IUCN status data are aggregated in a non-traditional way, with NT combined with threatened categories rather than with LC as non-threatened (the IUCN considers NT as non-threatened). The authors should justify their grouping. Also while I realize sample size and unbalanced groups may be a problem, MCMCglmm allows for fitting multinomial model which better reflect the status data.

The authors talk about differences based on a species' ecology and life-history but only considered two aspects: whether a species breeds on land or ice, and SSD. The text could be revised to reflect that narrow scope (and ideally explain why they focus on only those two traits) or even better, the authors could consider other traits to make a stronger case on the importance of traits. There are potentially many other relevant ecological and life-history traits such as reproductive rates, generation time – both linked to how fast pop size may recover; social structures and breeding seasonality – linked to how easy to hunt/exploit, geographic area – regions in which exploitation may have been unlikely.

Related to this point, land breeding was more likely to be associated with bottleneck signals, and this is explained by accessibility and pop density. I was left wondering if those two proposed mechanisms could not in fact be tested? For instance there is an ice-breeding seal with high P_{bot} , is this distinct from other ice-breeders in terms of accessibility or pop density?

Line 324 "We also found that heterozygosity-excess was strongly linked to sexual size dimorphism (SSD)" Does the relationship change if you exclude the Southern elephant seal?

Are results influenced by the number of loci or individuals sampled?

Line 500 "Details of all the models are given in the supplementary material" presumably this refers to supplementary table 8. The legend of that table reads "Estimated parameters for the main Bayesian phylogenetic mixed models" which makes me wonder if there are other non-main models tested.

Line 505 "For all models, we report the marginal R^2 as in 66" Is there a reason not to report the conditional R^2 too? It may be interesting to know how much of the total variance the full model explained.

Line 398 "Overall, 6% of loci were found to deviate from HWE in both tests and as these are unlikely to affect our comparative analyses, we focused subsequently on the full datasets." Why are these unlikely to affect the analyses?

Reference 13 does not list authors.

Line 396 the reference has a unedited format

Reviewer #2 (Remarks to the Author):

Decision: accept conditional on major revisions

The manuscript "Recent demographic histories and genetic diversity across pinnipeds are shaped by anthropogenic interactions and mediated by ecology and life-history" by Stoffel et al. is a very nice study that showed that land breeders had a more drastic demographic history impacted by hunting than ice-breeders. There are also some other very cool basic results relating allelic richness with global abundance and ABC bottleneck model probability. I am not confident of the results for various reasons stated below, but I think that some more work can remedy this.

Major Points

The abstract details some of the data yet only says "genetic data". It is crucial to provide more details in the abstract about what type of genomic sampling took place.

The motivation of the comparative approach is strong, but the authors ought to cite some of the work already done with ABC, especially the studies focusing on bottleneck expansion models (e.g. Xue and Hickerson, Burbrink et al, Gehara et al.). The authors also ought to cite the 90s (?) Roman/Palumbi-related papers examining the weaker than predicted impact that whaling had on observed genetic diversities.

To strengthen the inferential confidence, the authors should consider a model that also incorporates a crucial features that could have a major impact. Late Pleistocene history bottleneck expansions of huge magnitude associated with the LGM could have had a huge impact on the data, even if there was a subsequent one in the 1800s. I liked that the authors repeated the analysis on the largest clusters (informed by STRUCTURE results of $K > 2$).

The casual mention of "at selectively neutral loci such as microsatellites is an indicator of recent bottlenecks because during a population decline the number of alleles decreases faster than heterozygosity (Nei et al. 1975)" is difficult to digest with a straight face in light of the well known strong possibility of strong linkage to parts of the genome undergoing positive selection. The authors ought to loudly acknowledge that their bottleneck expansion inference could have been confounded by this process.

Why are bottleneck histories mutually exclusive with historical population expansions? It makes sense that they could often occur in sequence (expansion followed by a bottleneck). The way it is stated it sounds like the ABC model did not allow for both to happen.

The result that the majority of species fit a no-growth model over the bottleneck model makes me think that major point about the LGM above could have confounded the results.

The authors report a "good fit to the data" but did the authors conduct posterior predictive tests (as recommended best ABC practices to verify that the models could largely generate the observed data)? I see in supp table 4A the authors report "prediction error" being good, but it is hard to gain a intuition here. To gain the confidence of the reader, the authors need to just produce simple dot plots from the "leave one out" cross validation (i.e. plot real values vs the point estimates such as the mode estimates).

Minor points

By "neutral model" do the authors really mean the null no-growth model? The Bottleneck model also assumes neutrality.

Where are the STRUCTURE results? I don't see any table or figure cited.

Response to referees, *Stoffel et al.*

Reviewer #1 (Remarks to the Author):

The ms “Recent demographic histories and genetic diversity across pinnipeds are shaped by anthropogenic interactions and mediated by ecology and life-history” evaluates the evidence for genetic bottlenecks in multiple pinniped populations worldwide and tests how genetic diversity and bottleneck signal correlate with some other factors. The authors finds some evidence of genetic bottlenecks but not to the extent expected given the historical exploitation of many pinnipeds.

This is a well-written ms and I commend the reviewers for making the data and R script available. I have a few specific comments and questions which I detail below.

Many thanks for the positive comments on the writing and presentation of data and code. We place high value on the transparency and reproducibility of our analyses.

Regarding the genetic data analysed, the authors do not explain how the relevant published microsatellite data were identified or selected from the available information. I know of at least 1 source which is not included and that provides data for one of the studied species in different sampling areas (see citation below). The authors need to explain how they identified sources and, if relevant, why some were excluded (and whether this could affect results). This is important to be able to understand how representative the results may be. González-Suárez, M; Aurióles-Gamboa, D; Gerber LR (2010) Past exploitation of California sea lions did not lead to a genetic bottleneck in the Gulf of California *Ciencias Marinas* 36: 199-211. Note that the data for this study are freely available at Dryad Digital Repository doi:10.5061/dryad.1454.

Response 1 Thanks for raising this point. We did indeed conducted systematic searches to identify suitable datasets and we now report on this in a new section at the beginning of Supplementary Information part 3 at p.28, which includes a new table of our literature survey results. Briefly, we conducted Web of Science searches (last updated 28th June 2018) separately for all 35 species, combining the term 'microsat*' with all known latin and common species names. This led to the identification of a total of 304 unique records (see Supplementary information 1, Supplementary Table 1, or below). For each species, we then identified one or a small number of papers reporting datasets that were deemed suitable on this basis of the balance between the number of loci and individuals. As in most cases the raw data were not publically available, we contacted the authors of what we considered to be the most appropriate dataset for each species for our analyses. This allowed us to collate a single high quality microsatellite dataset for each of 25 pinniped species.

For several species, we could not identify any suitable records, and this is what motivated us to generate new microsatellite datasets for five further species where samples could be obtained. For many more species, only a single appropriate dataset could be identified. Although for a few species multiple datasets are in principle available, we feel our approach of selecting the single most appropriate dataset is valid for the following reasons:

(i) Our selected datasets are the largest available to our knowledge for each species with regard to the number of overall genotypes. We are aware of the study of California sea lions highlighted by the referee, but the dataset in question is smaller than the dataset used in our study (3550 versus 4511 genotypes respectively).

(ii) Merging microsatellite datasets for the same species from multiple studies is not possible as differences in loci, genotyping and scoring methodologies prohibit combining these datasets in a meaningful way.

(iii) Analysing multiple datasets for a small subset of species would introduce a pseudoreplication problem into most of our analysis.

(iv) We show that our analyses are not sensitive to population structure (see Supplementary Fig.6). By implication, adding a few additional datasets is unlikely to change our results as well.

Scientific name	Common name	Web of Science search term	Results (n = 304)
Odobenus rosmarus rosmarus	Walrus	("Odobenus rosmarus rosmarus" OR "Walrus") AND microsat*	15
Callorhinus ursinus	Northern Fur Seal	("Callorhinus ursinus" OR "Northern Fur Seal") AND microsat*	4
Neophoca cinerea	Australian Sea Lion	("Neophoca cinerea" OR "Australian Sea Lion") AND microsat*	2
Otaria flavescens	South American Sea Lion	("Otaria flavescens" OR "Otaria byronia" OR "South American Sea Lion") AND microsat*	4
Arctocephalus pusillus doriferus	South African Fur Seal	("Arctocephalus pusillus doriferus" OR "South African Fur Seal" OR "Australian Fur Seal") AND microsat*	1
Phocartos hookeri	Hooker's Sea Lion	("Phocartos hookeri" OR "Hooker's Sea Lion" OR "New Zealand Sea Lion") AND microsat*	9
Arctocephalus forsteri	New Zealand Fur Seal	("Arctocephalus forsteri" OR "New Zealand Fur Seal") AND microsat*	6
Arctocephalus australis	South American Fur Seal	("Arctocephalus australis" OR "South American Fur Seal") AND microsat*	6
Arctocephalus galapagoensis	Galapagos Fur Seal	("Arctocephalus galapagoensis" OR "Galapagos Fur Seal") AND microsat*	2
Arctocephalus gazella	Antarctic Fur Seal	("Arctocephalus gazella" OR "Antarctic Fur Seal") AND microsat*	49
Arctocephalus tropicalis	SubAntarctic Fur Seal	("Arctocephalus tropicalis" OR "SubAntarctic Fur Seal") AND microsat*	2
Arctocephalus philippii	Juan Fernandez Fur Seal	("Arctocephalus philippii" OR "Juan Fernandez Fur Seal") AND microsat*	0
Arctocephalus townsendi	Guadalupe Fur Seal	("Arctocephalus townsendi" OR "Guadalupe Fur Seal") AND microsat*	0
Eumetopias jubatus	Steller's Sea Lion	("Eumetopias jubatus" OR "Steller's Sea Lion" OR "Steller Sea Lion") AND microsat*	15
Zalophus californianus	California Sea Lion	("Zalophus californianus" OR "California Sea Lion") AND microsat*	33
Zalophus wollebacki	Galapagos Sea Lion	("Zalophus wollebacki" OR "Galapagos Sea Lion") AND microsat*	10
Erigonathus barbatus	Bearded seal	("Erigonathus barbatus" OR "Bearded seal") AND microsat*	1
Cystophora cristata	Hooded seal	("Cystophora cristata" OR "Hooded seal") AND microsat*	3
Phoca hispida	Ringed seal	("Phoca hispida" OR "Ringed seal") AND microsat*	13
Phoca sibirica	Baikal seal	("Phoca sibirica" OR "Baikal seal") AND microsat*	0
Halichoerus grypus	Grey seal	("Halichoerus grypus" OR "Grey seal") AND microsat*	52
Phoca caspica	Caspian seal	("Phoca caspica" OR "Caspian seal") AND microsat*	0
Phoca largha	Spotted seal	("Phoca largha" OR "Spotted seal" OR "Largha seal") AND microsat*	2
Phoca vitulina vitulina	Harbour seal	("Phoca vitulina vitulina" OR "Harbour seal") AND microsat*	23
Phoca fasciata	Ribbon seal	("Phoca fasciata" OR "Histriophoca fasciata" OR "Ribbon seal") AND microsat*	0
Phoca groenlandica	Harp seal	("Phoca groenlandica" OR "Harp seal") AND microsat*	4
Lobodon carcinophagus	Crabeater seal	("Lobodon carcinophagus" OR "Crabeater seal") AND microsat*	3
Ommatophoca rossi	Ross seal	("Ommatophoca rossi" OR "Ross seal") AND microsat*	3
Hydrurga leptonyx	Leopard seal	("Hydrurga leptonyx" OR "Leopard seal") AND microsat*	4
Leptonychotes weddelli	Weddell seal	("Leptonychotes weddelli" OR "Weddell seal") AND microsat*	6
Mirounga angustirostris	Northern Elephant seal	("Mirounga angustirostris" OR "Northern Elephant seal") AND microsat*	8
Mirounga leonina	Southern Elephant seal	("Mirounga leonina" OR "Southern Elephant seal") AND microsat*	9
Monachus monachus	Mediterranean Monk seal	("Monachus monachus" OR "Mediterranean Monk seal") AND microsat*	4
Monachus schauinslandi	Hawaiian Monk seal	("Monachus schauinslandi" OR "Hawaiian Monk seal") AND microsat*	8
Monachus tropicalis	Caribbean Monk seal	("Monachus tropicalis" OR "Caribbean Monk seal") AND microsat*	0
Pusa hispida saimensis	Saimaa ringed seal	("Pusa hispida saimensis" OR "Saimaa ringed seal") AND microsat*	3
Pusa hispida ladogensis	Ladoga ringed seal	("Pusa hispida ladogensis" OR "Ladoga ringed seal") AND microsat*	0
Pusa hispida botnica	Baltic ringed seal	("Pusa hispida botnica" OR "Baltic ringed seal") AND microsat*	0

Supplementary Table 12: Identification of microsatellite datasets. We searched relevant papers using scientific names and common names of each species, as shown in the "Web of Science search term" column. The "Results" column shows the number of papers found with the respective search term.

Related to this issue, I was unclear on how many sampling sites were represented in each dataset. This is important to understand how representative the data are and also key to better understand the STRUCTURE analyses.

Response 2 We have added the number of sampling locations to Supplementary Table 1. The names of the sampling sites are provided together with the raw genotypes and will be deposited on Dryad upon acceptance of the article. This information is also available via https://github.com/mastoffel/pinniped_bottlenecks/blob/master/data/processed/seal_data_largest_clust_and_pop_30.xlsx. Furthermore, documented scripts are available in the data processing pipeline to easily retrieve and investigate the data further.

For several species the posterior probabilities for the bottleneck and neutral models are very similar (eg Walrus, Baltic ringed seal, NZ sea lion, and others). Given the posterior probability of correct model classification are relatively low (error rates of 25-30%) I am not convinced an arbitrary threshold of 0.5 can be used as evidence of fitting a bottleneck vs neutral model. Wouldn't the rejection of the neutral model be a better way to determine which species show patterns consistent with a bottleneck? Using that approach one can clearly identify the four species for which bottleneck N_e is clearly estimated as small in figure 2.

Response 3 It is common and accepted practice in ABC modeling to estimate parameters under the model with the highest posterior probability. We therefore feel that 0.5 is not an arbitrary threshold. Moreover, by selecting the model with the highest probability for each species, we essentially reject the model with the lower probability. We are therefore not sure what the reviewer means by their suggestion of 'rejecting the neutral model'.

The referee is correct in pointing out from Figure 2 that four species of pinniped were strongly bottlenecked. However, we know from history of sealing that many other pinniped species were severely hunted. This is

reflected not only by the four most extreme examples, but also in the bottleneck effective population sizes of the seven further species shown in Figure 2. We therefore feel that the inclusion of these species is justified based on the ABC analysis and reflects what we know very well from the history of sealing.

Importantly, our model selection only affects the list of species for which we estimate bottleneck effective population sizes. What is not affected is all of the downstream phylogenetic mixed models on which most of our inferences are based. These models all analyse the bottleneck model probability (p_{bot}), which is a continuously distributed variable and does not rely on a threshold.

Finally, in response to this comment as well as the comment below (Response 4), we made major changes to our manuscript by defining a more extreme bottleneck scenario and re-running all of the downstream analyses. This resulted in significantly improved model classification, but our results and thus our conclusions did not change. The same species favoured the bottleneck scenario, even after substantial changes to the analysis, suggesting that our model selection is robust.

Related to this in line 197 “ABC was clearly able to distinguish between the two models, with the posterior probability of correct model classification being 75% for the bottleneck model and 71% for the neutral model” I may be misinterpreting these results, but those probabilities of correct classification do not seem particularly good, you can be wrong about 25-30% of the times, which is not trivial.

Response 4 The reviewer is right in saying that our model classification probabilities could be improved upon. However, our ABC analysis is unique in that we simulated genetic data based on just two scenarios which had to broadly fit to 30 different pinniped species. This required broad priors for both models, which unavoidably results in a certain amount of overlap between the models when trying to classify a randomly chosen simulation. Hence, some degree of misclassification is inevitable when comparing realistic scenarios that should capture a broad range of demographic histories.

To tackle this comment, we have now done a major re-analysis of our data (see also Response 3). First of all, we defined the two scenarios more distinctly by specifying tighter priors on the bottleneck effective population size ($N_{\text{e,bot}}$) under the bottleneck scenario. $N_{\text{e,bot}}$ now ranges from one to 500 individuals as opposed to one to 800 in the previous analysis. This resulted in a substantially improved model classification, with simulations under the bottleneck model being correctly classified 85% of the time and simulations under the neutral model being correctly classified 89% of the time.

Based on the newly derived model probabilities, we re-estimated all posterior distributions and re-ran all of the downstream phylogenetic mixed models. While the models are now more distinct and thus better classifiable, all downstream inference remains unaltered.

We have now modified the methods and results section (lines 198-203) and all of the figures to incorporate the results of the new analysis. For visualisation purposes, we also now present the results of the model classification analysis as a confusion matrix, which is shown below and is now given as Supplementary Fig. 1.

Supplementary Fig. 1: Confusion matrix plot showing the misclassification rate estimates from our model selection procedure (see Methods for details). Simulations under the bottleneck model are shown in dark grey and simulations generated under the neutral model are shown in light grey. The two bars show the classification of the simulations into either the bottleneck or the neutral model. When a simulation was randomly chosen from the bottleneck model, it was classified into the bottleneck model 85% of the time. When a simulation was randomly chosen from the neutral model, it was classified into the neutral model 89% of the time.

Figure 1 (and in fact all figures) is visually appealing but I find it difficult to see the probability values with the colors. Using the actual values would be much more informative and given the colours are already in a table format, that seems very much doable.

Response 5 Many thanks for complimenting us on the quality and visual appeal of the figures. However, we don't agree that Figure 1 would be improved by replacing the heat map element with actual numbers. We think the overall patterns are visually clear, as we used an established colour palette which is designed to maximise differences between the diverging ends of a scale. In addition, the values themselves are easily accessible in Supplementary Tables 2 and 3, which is clearly stated in the figure legend.

Line 199 "for every species the preferred model showed a good fit to the data" for the Saimaa ringed seal the p-value is 0.08, perhaps works discussing a bit more uncertainty in this case?

Response 6 Similar to the northern elephant seal, the Saiima ringed seal shows and extremely low genetic diversity. This causes both species to be on the lower end of the spectrum of simulated diversity under the bottleneck model and hence the p-values are towards the lower end of those observed across pinnipeds. This is a consequence of having to specify broad priors to fit a large range of species. However, the p-value of the Saiima ringed seal has now increased to 0.14 based on our new analysis (see Responses 3 and 4) which restricted the $N_{e,bot}$ to between one and 500 individuals. The simulated genetic diversity is therefore on average lower and fits better to species at that end of the spectrum such as the Saiima ringed seal and northern elephant seal.

IUCN status data are aggregated in a non-traditional way, with NT combined with threatened categories rather than with LC as non-threatened (the IUCN considers NT as non-threatened). The authors should justify their grouping. Also while I realize sample size and unbalanced groups may be a problem, MCMCglmm allows for fitting multinomial model which better reflect the status data.

Response 7 We agree with this criticism, which we have now addressed by changing the IUCN status categories according to the reviewer's suggestion. We now grouped *least concern* and *near threatened* into a '*low concern*' and *vulnerable* and *endangered* into a '*high concern*' group. We then re-ran the MCMCglmm analysis for these revised groupings and the results did not change appreciably. We've updated the results section in-

cluding Figure 5 accordingly (lines 323-331).

We also concur that given a large enough dataset it would be desirable to analyse all four categories (i.e. *least concern*, *near threatened*, *vulnerable* and *endangered*) separately rather than grouping them into two main categories, as this might better reflect the status data. The reviewer is correct in saying that, at least in principle, this could be done using a multinomial model in MCMCglmm. However, as also recognised by the reviewer, our sample size of species is too small for this to work, especially because the *near threatened* category contains only species, the *vulnerable* category contains only four species and the *endangered* category contains only six species.

The authors talk about differences based on a species' ecology and life-history but only considered two aspects: whether a species breeds on land or ice, and SSD. The text could be revised to reflect that narrow scope (and ideally explain why they focus on only those two traits) or even better, the authors could consider other traits to make a stronger case on the importance of traits. There are potentially many other relevant ecological and life-history traits such as reproductive rates, generation time – both linked to how fast pop size may recover; social structures and breeding seasonality – linked to how easy to hunt/exploit, geographic area – regions in which exploitation may have been unlikely.

Response 8 We agree, and have therefore followed the reviewer's suggestion of incorporating additional variables into this analysis. However, we had to work within some constraints. First of all, data are not available for all of the traits that we would ideally like to analyse. In particular, data on reproductive rate are lacking for the vast majority of species, as this requires long-term individual based studies quantifying lifetime reproductive success. Second, the complexity of our models is limited due to the sample size of 30 species. In other words, our models will not converge if we add many more predictors.

To address this comment, we have incorporated two additional variables that we believe are relevant and for which data are available for all of the species. The first of these, as suggested by the reviewer, is generation time, as species with short generation times should be able to recover more quickly from bottlenecks. The second variable we chose to incorporate was the length of the breeding season, as we reasoned that species with long breeding seasons could be more prone to exploitation as sealers have a larger time window in which to hunt the animals. We think this is broadly analogous to 'breeding seasonality'. With regards to 'social structures', we're not really sure what the reviewer means by this. Detailed data on social organisation, patterns of genetic relatedness etc., are not available for the vast majority of species. However, we believe that SSD captures the most important social structures across pinnipeds—their mating and harem systems.

We incorporated these additional variables into the Bayesian phylogenetic mixed models of $p_{\text{het-exc}}$ and p_{bot} . Our results for breeding habitat and SSD remain unchanged and neither breeding season length nor generation time further impacted bottleneck strength across pinnipeds. We re-wrote the corresponding results section (lines 281 – 303) and revised Fig. 3, which is also shown below.

We would also have liked to include these two variables in our model of genetic diversity. However, the existing model already contains five predictor variables. We therefore fitted a separate model of A_r containing breeding season length and generation time as predictor variables. Neither variable was related to A_r (breeding season: $\beta = 0.01$ CI [-0.03, 0.01], generation time: $\beta = 0.00$ CI [0.00, 0.01]). Consequently, we have not altered the presentation of our original model although we have added a statement to the effect that our new variables are not associated with A_r . (lines 308 – 311)

We are grateful to the reviewer for this comment and we believe that our new analyses broadens the scope of the paper and justify our general use of the terms ecology and life history.

Fig. 3: Ecological and life-history effects on bottleneck signatures. Shown are the results of phylogenetic mixed models of $prop_{het-exc}$ and p_{bot} with breeding habitat and SSD fitted as fixed effects. Panels A and B show differences between ice- and land-breeding species in $prop_{het-exc}$ and p_{bot} respectively. Raw data points are shown together with standard Tukey box plots. Panel C shows the relationship between sexual size dimorphism (SSD) and $prop_{het-exc}$ with individual points colour coded according to the ABC bottleneck probability (p_{bot}) and the line representing the predicted response from the $prop_{het-exc}$ model. Marginal and unique R^2 values, standardized β coefficients and structure coefficients are shown for models of $prop_{het-exc}$ (filled points) and p_{bot} (open points) in panels D–F, where they are presented as posterior medians with 95% credible intervals. Species abbreviations are given in Fig. 1 and Supplementary Table 1.

Related to this point, land breeding was more likely to be associated with bottleneck signals, and this is explained by accessibility and pop density. I was left wondering if those two proposed mechanisms could not in fact be tested? For instance there is an ice-breeding seal with high p_{bot} , is this distinct from other ice-breeders in terms of accessibility or pop density?

Response 9 We agree that the two most accessible ice-breeding seals (the Ladoga ringed seal and Saimaa ringed sea) indeed show the strongest bottleneck signals and the lowest diversities among ice-breeding seals. However, this could also be a reflection of limited geographic ranges.

We used the terms 'accessibility' and 'density' rather loosely to refer to broad-scale differences between ice and land-breeding seals. The majority of ice breeding seals, especially polar species such as Weddell and crabeater seals, breed in remote areas that will usually be inaccessible by ship due to the presence of dense pack ice. By contrast, land-breeding seals tend to be found in high densities on breeding beaches that could be easily accessed by sealers.

It would of course be nice to try to tease apart the relative contributions of accessibility and density, but unfortunately quantitative data on both aspects are lacking, even for the most intensively studied species. We therefore used breeding habitat and SSD as proxies. These variables are not only readily quantifiable across all species but also explain a large proportion of the variance in our data.

Line 324 “We also found that heterozygosity-excess was strongly linked to sexual size dimorphism (SSD)” Does

the relationship change if you exclude the Southern elephant seal?

Response 10 We repeated the analysis of SSD after excluding the Southern elephant seal and found that both R^2 and β were highly similar (Before exclusion: $R^2 = 0.30$, CI [0.00, 0.64], stand. $\beta = 0.07$, CI [0.00, 0.13], after exclusion: $R^2 = 0.26$, CI [0.00, 0.79], stand. $\beta = 0.08$, CI [0.00, 0.17]). We now state this clearly in the corresponding section of the results section and added an additional figure to the Supplementary material.

Supplementary Fig. 8: Robustness of the relationship between $\text{prop}_{\text{het-exc}}$ and SSD to the exclusion of the southern elephant seal (SES). The left panel shows the raw data and model prediction for the full dataset, while the right panel presents equivalent results for the dataset excluding the southern elephant seal.

Are results influenced by the number of loci or individuals sampled?

Response 11 Our results are not influenced by the sample size either of individuals or loci for the following reasons:

- (1) The number of individuals has already been controlled for in all of our analyses. Specifically, we used a thorough standardization procedure to eliminate the effects of sample size variation on genetic diversity by randomly sub-sampling ten individuals from each dataset 1000 times with replacement and calculating the corresponding mean and 95% confidence interval for each summary statistic. These standardized genetic summary statistics were used in all analyses.
- (2) The number of loci will not systematically bias the mean of any summary statistic as all of them are calculated as a mean across loci. Therefore, the expected mean will be the same independently of how many loci are used.

In response to this comment, we have clarified in both the methods and results sections how we standardised our summary statistics over individuals. Furthermore, we have included a statement in the methods section (lines 463-468) outlining why differences in the number of loci will not bias our analyses.

Line 500 “Details of all the models are given in the supplementary material” presumably this refers to supplementary table 8. The legend of that table reads “Estimated parameters for the main Bayesian phylogenetic mixed models” which makes me wonder if there are other non-main models tested.

Response 12 Apologies for the confusion. We did not construct any other models and so we have re-phrased the table legend to clarify this point.

Line 505 “For all models, we report the marginal R^2 as in 66” Is there a reason not to report the conditional R^2 too? It may be interesting to know how much of the total variance the full model explained.

Response 13 We originally reported only marginal R^2 values because we were interested in the strength of the fixed effects after controlling for phylogenetic relatedness. By contrast, the conditional R^2 also includes variation attributable to the phylogeny. To address this comment, we have included conditional R^2 values for all of our models in Supplementary Table 10. Phylogenetic effects are small in all of our models and our conclusions remain unchanged.

Line 398 “Overall, 6% of loci were found to deviate from HWE in both tests and as these are unlikely to affect our comparative analyses, we focused subsequently on the full datasets.” Why are these unlikely to affect the analyses?

Response 14 We conducted broad analyses including a large number of loci across many different species. Excluding a very small proportion of loci from these datasets is unlikely to appreciably change the means of the calculated summary statistics and consequently our downstream analyses.

Nevertheless, we have now repeated the analyses shown in Figure 1 after excluding loci from each dataset that deviated significantly from Hardy-Weinberg equilibrium. The results, shown in the figure below, are essentially unchanged (all repeatabilities > 0.95). We have updated the manuscript to include a new results section containing this analysis (lines 237 – 242), and a new supplementary figure and table (Supplementary Fig. 7, Supplementary Table 8).

Supplementary Fig. 7: Replicated Figure 1 based on reduced datasets containing only loci in Hardy-Weinberg equilibrium for each dataset.

Reference 13 does not list authors.

Response 15 We have corrected the reference, thanks for spotting this.

Line 396 the reference has a unedited format

Response 16 We also corrected this reference, thanks again.

Reviewer #2 (Remarks to the Author):

Decision: accept conditional on major revisions

The manuscript “Recent demographic histories and genetic diversity across pinnipeds are shaped by anthropogenic interactions and mediated by ecology and life-history” by Stoffel et al. is a very nice study that showed that land breeders had a more drastic demographic history impacted by hunting than ice-breeders. There are also some other very cool basic results relating allelic richness with global abundance and ABC bottleneck model probability. I am not confident of the results for various reasons stated below, but I think that some more work can remedy this.

Response 17 Many thanks for the positive comments and we have worked hard to incorporate all of these important points (see below).

Major Points

The abstract details some of the data yet only says “genetic data”. It is crucial to provide more details in the abstract about what type of genomic sampling took place.

Response 18 We have changed the abstract to reflect the fact that our genetic data are from microsatellites.

The motivation of the comparative approach is strong, but the authors ought to cite some of the work already done with ABC, especially the studies focusing on bottleneck expansion models (e.g. Xue and Hickerson, Burbrink et al, Gehara et al.). The authors also ought to cite the 90s (?) Roman/Palumbi-related papers examining the weaker than predicted impact that whaling had on observed genetic diversities.

Response 19 We already cited over ten papers including both empirical and methodological ABC studies. Nevertheless, we have added several new references to the introduction section in order to strengthen our case for using ABC (see below). We have also cited the Roman and Palumbi paper in the introduction and the Burbrink paper in a new results section dealing with postglacial expansions (see Responses 20). However, in some of the above cases we were not sure which paper the referee was referring to. Therefore, we would be happy to incorporate further papers given more detailed guidance.

Papers newly cited:

- (i) Chan, Y. L., Anderson, C. N., & Hadly, E. A. (2006). Bayesian estimation of the timing and severity of a population bottleneck from ancient DNA. *PLoS Genetics*, 2(4), e59.
- (ii) Xue, A. T., & Hickerson, M. J. (2015). The aggregate site frequency spectrum for comparative population genomic inference. *Molecular ecology*, 24(24), 6223-6240.
- (iii) Duchon, P., Živković, D., Hutter, S., Stephan, W., & Laurent, S. (2013). Demographic inference reveals African and European admixture in the North American *Drosophila melanogaster* population. *Genetics*, 193(1), 291-301.
- (iv) Roman, J., & Palumbi, S. R. (2003). Whales before whaling in the North Atlantic. *science*, 301(5632), 508-510.
- (v) Burbrink, F. T., Fontanella, F., Pyron, R. A., Guirer, T. J., & Jimenez, C. (2008). Phylogeography across a continent: the evolutionary and demographic history of the North American racer (Serpentes: Colubridae: *Coluber constrictor*). *Molecular Phylogenetics and Evolution*, 47(1), 274-288.

To strengthen the inferential confidence, the authors should consider a model that also incorporates a crucial

features that could have a major impact. Late Pleistocene history bottleneck expansions of huge magnitude associated with the LGM could have had a huge impact on the data, even if there was a subsequent one in the 1800s.

Response 20 We think it's important to strongly focus our study on recent bottlenecks for several reasons. First, our study is based on a very clear hypothesis: sealing has severely impacted the population sizes and genetic diversity of pinnipeds. This hypothesis is founded on a large body of literature reporting severe reductions in pinniped population sizes during the 18th and 19th centuries. Second, this focus on a short time-frame and well known sealing history allows us to clearly define our models around reasonable and rather tight priors. Third, the focus on recent demographic declines also allows us to compare our ABC results to a widely used and established method - heterozygosity excess, which is a different but complimentary approach to quantify recent population size changes. Fourth, Hoban et al¹ have shown that bottlenecks followed by an extended period of recovery (i.e. many hundreds of generations after the last glacial maximum, LGM) are difficult to detect using microsatellites. Therefore we do not believe our microsatellite data are well suited to modelling events further back in a species' history.

Of course it would be interesting to model post-glacial expansions. However, we cannot frame such an analysis around a clear hypothesis as virtually nothing is known about how different pinniped species were affected by changes in habitat availability during the last ice age. Furthermore, in contrast to our analysis of recent bottlenecks, it is difficult to define appropriate priors given this almost complete lack of knowledge.

Finally, our ABC analysis is unique in that we defined two models to broadly capture the demographic histories of 30 species. We believe that such a wide-reaching analysis has to be kept simple to facilitate comparative analyses. Using the bottleneck probability (p_{bot}) from our ABC analysis, we found strong and expected associations with both breeding habitat and allelic richness. This shows that our models, although simple, capture meaningful variation in recent demographic histories across pinnipeds.

To address the reviewer's concerns, however, we have conducted a major new analysis of the data. Following the reviewer's suggestion, we evaluated whether small population sizes during the LGM followed by expansion could cause similar genetic patterns across pinnipeds to recent bottlenecks caused by anthropogenic exploitation. Specifically, we simulated four scenarios, which included the two scenarios from the main analysis ('bottleneck' and 'neutral' models) plus two scenarios that were identical to the former two but which also included a smaller population size during the LGM followed by an expansion ('LGM + bottleneck' and 'LGM + neutral' models). In contrast to the main analysis in which we specified just two models, ABC could not reliably distinguish between these four models. Correct rates of model classification were 64% for the bottleneck model, 60% for the LGM + bottleneck model, 60% for the neutral model and 66% for the LGM + neutral model. This suggests that ABC cannot reliably distinguish between broadly equivalent models that do and do not include ice age effects. However, none of the 11 species that supported the bottleneck model in our main analysis were found to support the LGM + neutral model in our new analysis. This suggests that genetic patterns in our dataset caused by recent bottlenecks are different from those expected under a postglacial expansion model. Furthermore, in our new analysis, all 11 species that originally supported the bottleneck model again had the highest posterior probability for one of the two scenarios incorporating a recent bottleneck. Eight of these still supported the more simple bottleneck model. Consequently, our inference of recent bottlenecks remains unaltered regardless of whether or not these were preceded by an ice age effect.

We believe our study is strengthened by this new analysis. However, for the reasons described above, as well as our empirical findings of much greater uncertainty in model classification as well as a lack of evidence that our inference of recent bottlenecks is confounded by ice age effects, we do not believe that this analysis is preferable to our two-model ABC analysis. We therefore extensively report this analysis in the Supplementary Information (p.22 - 27) as well as in a new results paragraph of the main manuscript (lines 233 - 263).

Finally, in response to this and several other comments by the reviewer, we have also expanded our discussion section (lines 350-355, lines 370-385) to further clarify and justify our approach, as well as to discuss the re-

sults of our new analyses which we believe strengthen our conclusions.

I liked that the authors repeated the analysis on the largest clusters (informed by STRUCTURE results of $K > 2$).

Response 21 Great. We believe that this analysis, together with the new re-analysis of the data after Hardy-Weinberg filtering (see Response 14) and the new ABC analysis incorporating ice age effects (see Response 20) strengthen our conclusions.

The casual mention of “at selectively neutral loci such as microsatellites is an indicator of recent bottlenecks because during a population decline the number of alleles decreases faster than heterozygosity (Nei et al. 1975)” is difficult to digest with a straight face in light of the well known strong possibility of strong linkage to parts of the genome undergoing positive selection. The authors ought to loudly acknowledge that their bottleneck expansion inference could have been confounded by this process.

Response 22 We believe that selection is unlikely to explain our results given that we analysed averages across many loci and that selection would have to operate in the same direction across loci within species and across species to explain our comparative patterns. However, we have acknowledged this possibility in a new discussion paragraph detailing potential caveats (lines 381 – 385)

Why are bottleneck histories mutually exclusive with historical population expansions? It makes sense that they could often occur in sequence (expansion followed by a bottleneck). The way it is stated it sounds like the ABC model did not allow for both to happen.

Response 23 This appears to be a misunderstanding as it was not our intention to imply that recent bottlenecks are mutually exclusive with historical population expansions. Both our bottleneck and neutral models do in fact allow changes in population size from pre- to post-bottleneck. Consequently, in the case of neutral model, population expansion would be reflected as a smaller historical N_e relative to the current N_e , as the priors for both are independently drawn and fitted in the model. The bottleneck model similarly allows for long term changes in N_e while also incorporating a recent demographic reduction due to hunting. To make this clearer, we have now revised the corresponding methods and results sections (lines 491-503, lines 190 - 206) to provide a better explanation of how we defined our demographic scenarios.

The result that the majority of species fit a no-growth model over the bottleneck model makes me think that major point about the LGM above could have confounded the results.

Response 24 As explained in Response 23, neither the neutral model nor the bottleneck models are no-growth models. Furthermore, as explained in Response 20, our initial inference of recent bottlenecks in 11 species is not confounded by historically small population sizes and subsequent expansions.

The authors report a “good fit to the data” but did the authors conduct posterior predictive tests (as recommended best ABC practices to verify that the models could largely generate the observed data)? I see in supp table 4A the authors report “prediction error” being good, but it is hard to gain a intuition here. To gain the confidence of the reader, the authors need to just produce simple dot plots from the “leave one out” cross validation (i.e. plot real values vs the point estimates such as the mode estimates).

Response 25 We did indeed follow best ABC practices by carefully evaluating every step in our ABC analysis. This is described in detail in the Methods section (lines 547 – 570) and the results of these evaluations are given both in the main results section of the manuscript and in the Supplementary material (Supplementary Table 4 for the prediction errors of the model estimates and Supplementary Table 5 for the goodness of fit tests for the selected models across species.

The prediction errors from the cross validation are well below one, indicating that posterior estimates contain information about the underlying true parameter values, as described by Csillery et al. ². At the referee's re-

quest, we have included a scatterplot of true versus estimated values for the N_e bot. See below and Supplementary Fig. 3.

Supplementary Fig. 3: Scatter plot of the of the cross-validation evaluation of N_e bot. Shown are the true values plotted against the estimated values in our ABC analysis. The plot reflects a prediction error of 0.55 (see Supplementary Table 4).

Minor points

By “neutral model” do the authors really mean the null no-growth model? The Bottleneck model also assumes neutrality.

Response 26 Actually it was quite tricky to come up with a suitable name for the model not containing a bottleneck. We considered calling it a 'null model' but (i) this could easily be confused with a linear model containing no predictors, and (ii) this model is not a no-growth model (see Response 23). We therefore retained the name 'neutral model' but provide a clearer definition as well as a statement in the Methods to explain that we do not imply for either model a departure from the neutrality of genetic loci (lines 498-503).

Where are the STRUCTURE results? I don't see any table or figure cited.

Response 27 Good point! We have now summarised the STRUCTURE results in a new supplementary table (Supplementary Table 6) including the number of inferred genetic clusters and sample sizes corresponding to the largest clusters analysed. This table is clearly referred to in the text of the Results section now.

Reviewers' comments:

Reviewer #1 (Remarks to the Author):

The authors have carefully and satisfactorily addressed my comments on the previous version. I have no further suggestions or concerns.

Reviewer #2 (Remarks to the Author):

This is the second time I have reviewed the manuscript "Recent demographic histories and genetic diversity across pinnipeds are shaped by anthropogenic interactions and mediated by ecology and life-history" sent by Stoffel et al. to the same journal. I like the improvements they made, but there are some remaining issues to be resolved.

1.

First, maybe these confusions I was having with the "neutral model" are coming from the cartoon in Figure 7. The authors state that the neutral model does have size change, but the casual glance at the cartoon makes it look like a no growth model. I see that "Ne and Nehist" are labeled differently, but they look like the same size on the cartoon. If the model actually allows size change, this should be reflected in the cartoon. Importantly, the authors never say what type of growth is in the model. Is it gradual, exponential, or logistic? Is it instant size change at some time that is itself drawn from a prior? Can NeHist be larger or smaller than Ne in the neutral model? These would all have additional parameters that are drawn from priors and have to be indicated as such.

I gather that t-hist is the time of sudden change. If so, the cartoon should actually show a sudden change at that example t-hist to clarify. Accordingly, the model should be renamed the such and such size change model. The term "neutral model" just has too many connotations with the Kimura neutral model to be used safely and clearly.

2.

This lack of clarity consequently made it difficult for me to follow their description of their new exploration of the two additional models involving older demographic events at the scale of the LGM (I do however, commend the authors for exploring these two other models with new analyses). The authors should accordingly clarify their four cartoons in the Suppl materials. Despite what the authors say about microsatellite data having little info on the LGM, I think the results of the classification tests are the four are really strong! (Supplementary Fig. 11). Furthermore, the new empirical model classifications are robust to there being a LGM expansion! (with regards to the 11 bottleneck species). I would really highlight this in the main text! These new results however need a little more unpacking. What about the non-bottleneck species? Do they also tend to fit either of the two non-bottleneck models? Maybe Supplementary Table 11 can be modified to indicate this. The authors could use bold to show the highest of the 4 posterior probabilities as well as add rows from the 2 model tests with the same bold numbers indicating the highest of two posterior probabilities.

A related point on this, all the excuses about priors the authors make about avoiding LGM models don't hold up well. Researchers build LGM models with ABC routinely with the same uncertainties, and the authors' new results show that it can work.

3.

I thank the authors for clarifying their use of prediction error and the new PODS plot showing an OK recovery of the bottleneck parameter with ABC estimation. But what about some kind of posterior or prior predictive tests? There are functions in ABC.r for this. I think this is considered standard ABC

best practices. At least show a PCA of the sumstats that are generated by drawing on the posterior accepted parameters (as well as the observed sumstat PCs).

4.

The citations I was referring to were papers that used ABC to specifically do a comparative analysis of population history across species (like the Burbrink et al. 2016 *ecol Letters* and Gehara et al. 2017 *Molec Ecol* papers). I think that Chan et al. 2014 *MBE* is the first example. Those other papers are great to cite as well.

5.

Linked selection could have likely gone in the same direction across all loci. In fact, in many of the heavily genomically sampled species, this appears to be the case (various flavors of positive selection and/or background selection). It's way beyond the scope of this study to really look into this, but the authors should not just lightly state that the loci are selectively neutral. It is good that the authors discuss this a little in the discussion.

Response to referees (2) , Stoffel et al.

Reviewer #1 (Remarks to the Author):

The authors have carefully and satisfactorily addressed my comments on the previous version. I have no further suggestions or concerns.

Reviewer #2 (Remarks to the Author):

This is the second time I have reviewed the manuscript “Recent demographic histories and genetic diversity across pinnipeds are shaped by anthropogenic interactions and mediated by ecology and life-history” sent by Stoffel et al. to the same journal. I like the improvements they made, but there are some remaining issues to be resolved.

Response 1. We are glad the referee likes the improvements and have endeavoured to incorporate the new comments.

1.

First, maybe these confusions I was having with the “neutral model” are coming from the cartoon in Figure 6. The authors state that the neutral model does have size change, but the casual glance at the cartoon makes it look like a no growth model. I see that “ N_e and N_{ehist} ” are labeled differently, but they look like the same size on the cartoon. If the model actually allows size change, this should be reflected in the cartoon.

Response 2. As the priors for N_e and N_{ehist} in the (formerly named) neutral model model are drawn independently from each other, this allows each species to vary in these two parameter estimates. Therefore, a certain species could be growing (modeled as instantaneous change) from N_{ehist} to N_e while another species could be declining and a third species could maintain a stable population. Given the 30 species analysed, there are a number of possibilities to what the non-bottleneck model could capture and therefore represent.

Although this was previously explained in the main text (lines 189-194, lines 490-501), we have now elaborated upon this further to improve clarity. First, we as requested, we have changed Figure 6. However, it's not obvious to us how to represent variation in the priors within a single figure, when in theory each simulation could generate a slightly different looking figure. We therefore modified the figure to include small figures of the prior distributions of each parameter, as these emphasise the potential variation captured across these parameters.

We have also expanded the figure legend to re-iterate the fact that pre-and post-bottleneck population sizes are allowed to vary for each species.

Importantly, the authors never say what type of growth is in the model. Is it gradual, exponential, or logistic? Is it instant size change at some time that is itself drawn from a prior? Can N_{ehist} be larger or smaller than N_e in the neutral model? These would all have additional parameters that are drawn from priors and have to be indicated as such.

Response 3. We modeled instantaneous population size changes drawn from pre-defined priors as described in the methods. As described above, the priors for N_{ehist} were drawn independently of the priors for N_e and are thus allowed to be smaller or larger. However, we added a sentence at the end of the respective Methods paragraph (lines 521-522) to further clarify this point.

I gather that t_{hist} is the time of sudden change. If so, the cartoon should actually show a sudden change at that example t_{hist} to clarify.

Response 4. The reviewer is right that t_{hist} is the time prior for an instantaneous population size change. However, both N_{hist} and N_e are drawn from the same prior distribution, and this is represented in the revised schematic of the two models. We believe it is not possible to depict the figure in any other way as for some species the sudden change might be growth and for others it might be decline.

Accordingly, the model should be renamed the such and such size change model. The term “neutral model” just has too many connotations with the Kimura neutral model to be used safely and clearly.

Response 5. We've taken this point on board and re-named the 'neutral model' as the 'non-bottleneck model'. We believe this is appropriate as the two models only differ in respect of whether a bottleneck was modeled. However, as explained above, both models are able to capture a certain degree of population size change but also a stable population. Therefore, we do not believe that it would be appropriate to re-name the neutral model a size-change model.

2.

This lack of clarity consequently made it difficult for me to follow their description of their new exploration of the two additional models involving older demographic events at the scale of the LGM (I do however, commend the authors for exploring these two other models with new analyses). The authors should accordingly clarify their four cartoons in the Suppl materials.

Response 6. We have clarified the schematic shown in Supplementary Figure 11 along the lines described in response 2. Specifically, we have added illustrations of the prior distributions of each parameter and expanded the model descriptions in the figure legend to re-iterate the detailed model description in the Supplementary Methods and to facilitate an easy overview.

Despite what the authors say about microsatellite data having little info on the LGM, I think the results of the classification tests are the four are really strong! (Supplementary Fig. 11).

Response 7. Although we appreciate the reviewer's enthusiasm, our rates of correct model classification are substantially lower than in the original analysis with two demographic scenarios, which is why we have to interpret the results with care. Nevertheless, we agree that we could have expanded upon these results further, and we have now done so (see Responses 8 and 9).

Furthermore, the new empirical model classifications are robust to there being a LGM expansion! (with regards to the 11 bottleneck species). I would really highlight this in the main text!

Response 8. We also feel this is important and supports our main argument and therefore we have clearly stated this observation in the results (lines 245-264) and discussion sections (lines 375-382) of the main manuscript as well in the corresponding supplementary results section (see response 9).

These new results however need a little more unpacking. What about the non-bottleneck species? Do they also tend to fit either of the two non-bottleneck models?

Response 9. We found a similar picture for the non-bottlenecked species, with 14 out of 19 species still supporting one of the two non-bottleneck models. The remaining five species were borderline in the original analysis (bottleneck model probability ≥ 0.4). These slightly different outcomes could potentially be due to lower model classification precision in the four-model analysis. However, they also highlight the importance of interpreting ABC results probabilistically. This is why we used the bottleneck model probability p_{bot} in our main analyses rather than a binary bottleneck / non-bottleneck variable. We have incorporated this as new paragraph to the respective supplementary results section.

Maybe Supplementary Table 11 can be modified to indicate this. The authors could use bold to show the highest of the 4 posterior probabilities as well as add rows from the 2 model tests with the same bold numbers indicating the highest of two posterior probabilities.

Response 10. We're afraid we don't really understand what the referee means by this but we now clearly state the main similarities and differences to the original analysis in the main text of the supplementary results.

A related point on this, all the excuses about priors the authors make about avoiding LGM models don't hold up well. Researchers build LGM models with ABC routinely with the same uncertainties, and the authors' new results show that it can work.

Response 11. We appreciated the discussion and believe the new analysis has substantially improved our manuscript and strengthened the robustness of our results.

3.

I thank the authors for clarifying their use of prediction error and the new PODS plot showing an OK recovery of the bottleneck parameter with ABC estimation. But what about some kind of posterior or prior predictive tests? There are functions in ABC.r for this. I think this is considered standard ABC best practices. At least show a PCA of the sumstats that are generated by drawing on the posterior accepted parameters (as well as the observed sumstat PCs).

Response 11. The goodness-fit-test which we report on (line 553-559, line 199-200) is considered a prior predictive test ¹, which we have clarified now (line 553). Posterior predictive tests are unfortunately not implemented in the abc package. However, we agree with the reviewer that these checks would be another way to evaluate model quality. We therefore followed the recommended procedure ^{2,3} of drawing 1,000 sets of parameters from the posterior distributions of each species and using those to re-simulate summary statistics across species. We have summarised these results in the form of histograms of each summary statistic rather than PCs, as we feel that this allows an easier evaluation of the posterior predictive checks. These are now provided as Supplementary figure 2, which is also shown below. The results indicate that preferred models for all species can reproduce the relevant observed summary statistics to a large degree. We have written an additional methods paragraph (lines 571-575) and refer to the new results in our revised results (lines 200-202) section of the main manuscript.

Supplementary Fig. 2: Posterior predictive checks for the summary statistics used in the ABC analysis (see Methods for details). After estimating posterior distributions of all parameters under the preferred model for each species (bottleneck versus non-bottleneck, colour coded orange and purple respectively), we sampled a set of 1,000 multivariate parameters from these distributions per species. Based on these parameters, we re-simulated data under the preferred model for each species to obtain 1,000 sets of summary statistics per species. The histograms show the distributions of these five summary statistics with the observed summary statistic of each species superimposed as a black vertical line. When interpreting these plots, it is important to bear in mind how informative a summary statistic can be for a given model. For example, the M-ratio is highly informative about recent bottlenecks and therefore showed good concordance between simulated and observed summary statistics for species supporting the bottleneck but not the non-bottleneck model.

4.

The citations I was referring to were papers that used ABC to specifically do a comparative analysis of population history across species (like the Burbrink et al. 2016 *ecol Letters* and Gehara et al. 2017 *Molec Ecol* papers). I think that Chan et al. 2014 *MBE* is the first example. Those other papers are great to cite as well.

Response 12. Thanks for the clarification. We now cite the Chan paper in the introduction and the Burbrink and Gehara papers in the section about post-glacial expansion.

5.

Linked selection could have likely gone in the same direction across all loci. In fact, in many of the heavily genomically sampled species, this appears to be the case (various flavors of positive selection and/or background selection). It's way beyond the scope of this study to really look into this, but the authors should not just lightly state that the loci are selectively neutral. It is good that the authors discuss this a little in the discussion.

Response 13. Thanks for that. We've taken this point on board and toned down our argument in the discussion.

Literature

1. Lemaire, L., Jay, F., Lee, I., Csilléry, K. & Blum, M. G. Goodness-of-fit statistics for approximate Bayesian computation. *arXiv preprint arXiv:1601.04096* (2016).
2. Gelman, A., Carlin, J. B., Stern, H. S. & Rubin, D. B. *Bayesian data analysis*. (Chapman and Hall/CRC, 1995).
3. Csilléry, K., Blum, M. G. B., Gaggiotti, O. E. & François, O. Approximate Bayesian Computation (ABC) in practice. *Trends in ecology & evolution* 25, 410–418 (2010).

REVIEWERS' COMMENTS:

Reviewer #2 (Remarks to the Author):

This is the third time i've reviewed this and am happy to see it's ready to fly. I still think the cartoon is slightly confusing in that it doesn't show a size change. I realize there is a chance that zero size change occurs, but with those priors, does it happen that often? If it were me, i would show two cases (size decline and size growth), as well as making a white line at this to delineate the two epochs.